# Synaptotagmin-1 C2B domain interacts simultaneously with SNAREs and membranes to promote membrane fusion

**Shen Wang[†], Yun Li[†], Cong Ma\***

Key Laboratory of Molecular Biophysics of the Ministry of Education, College of Life Science and Technology, Huazhong University of Science and Technology, Wuhan, China

**Abstract** Synaptotagmin-1 (Syt1) acts as a $Ca^{2+}$ sensor for neurotransmitter release through its C2 domains. It has been proposed that Syt1 promotes SNARE-dependent fusion mainly through its C2B domain, but the underlying mechanism is poorly understood. In this study, we show that the C2B domain interacts simultaneously with acidic membranes and SNARE complexes via the top $Ca^{2+}$-binding loops, the side polybasic patch, and the bottom face in response to $Ca^{2+}$. Disruption of the simultaneous interactions completely abrogates the triggering activity of the C2B domain in liposome fusion. We hypothesize that the simultaneous interactions endow the C2B domain with an ability to deform local membranes, and this membrane-deformation activity might underlie the functional significance of the Syt1 C2B domain in vivo.

**\*For correspondence:** cong.ma@ hust.edu.cn

[†]These authors contributed equally to this work

**Competing interests:** The authors declare that no competing interests exist.

## Introduction

$Ca^{2+}$-triggered neurotransmitter release by synaptic exocytosis is an exquisitely regulated process for interneuronal communication. The core machinery governing the process includes the SNAREs synaptobrevin, syntaxin-1 and SNAP-25, which form tight SNARE complexes to bridge synaptic vesicles to the plasma membrane and catalyze membrane fusion (*Jahn and Scheller, 2006*; *Sutton et al., 1998*; *Weber et al., 1998*). Syt1, the $Ca^{2+}$ sensor for the fast component of $Ca^{2+}$-triggered release (*Chapman, 2008*; *Fernandez-Chacon et al., 2001*; *Geppert et al., 1994*), confers $Ca^{2+}$ sensitivity to SNARE-dependent synaptic vesicle fusion. The triggering function of Syt1 depends on its interplay with membranes, SNAREs, complexins and other key proteins of the release machinery (*Rizo and Xu, 2015*).

Syt1 consists of two C2 domains, known as C2A and C2B. The C2A and C2B domains adopt similar structures and bind three and two $Ca^{2+}$ ions, respectively, through their $Ca^{2+}$-binding loops located at the top of the structures (*Fernandez et al., 2001*; *Shao et al., 1998*; *Sutton et al., 1995*). These loops mediate penetration of Syt1 C2 domains to acidic membranes containing phosphatidylserine (PS) in response to $Ca^{2+}$, and this activity is required for the triggering function of Syt1 (*Chapman and Davis, 1998*; *Fernandez-Chacon et al., 2001*; *Rhee et al., 2005*). In addition, Syt1 readily binds to phosphatidylinositol-4,5-bisphosphate (PI(4,5)P2) via its C2B domain in a $Ca^{2+}$-independent manner, which helps increase the apparent $Ca^{2+}$ affinity of Syt1 and thereby enhances release probability (*Bai et al., 2004*; *Li et al., 2006*; *Radhakrishnan et al., 2009*; *van den Bogaart et al., 2011a*). Furthermore, Syt1 binds to SNAP-25 and syntaxin-1, and to SNARE complexes mainly through its C2B domain, which is believed to position Syt1 on the pre-fusion SNARE complexes to trigger release in response to $Ca^{2+}$ (*Brewer et al., 2015*; *Zhou et al., 2015*; *de Wit et al., 2009*; *Mohrmann et al., 2013*).

**eLife digest** Information travels around the nervous system along cells called neurons, which communicate with each other via connections called synapses. When a signal travelling along one neuron reaches a synapse, it triggers the release of molecules known as neurotransmitters. These molecules are then taken up by the next neuron to pass the signal on. Neurotransmitters are stored in compartments called synaptic vesicles and their release from the first neuron depends on the synaptic vesicles fusing with the membrane that surrounds the cell. This "membrane fusion" process is driven by a group of proteins called the SNARE complex.

Membrane fusion is triggered by a sudden increase in the amount of calcium ions in the cell, which leads to an increase in the activity of a protein called synaptotagmin-1. A region of this protein known as the C2B domain is able to detect calcium ions, and it can also bind to the cell membrane and SNARE complex proteins. However, it is not clear what roles these interactions play in driving the release of neurotransmitters.

Wang, Li et al. have used a variety of biophysical techniques to study these interactions in more detail using purified proteins and other cell components. The experiments show that all three interactions occur at the same time and are all required for synaptotagmin-1 to trigger membrane fusion. Wang, Li et al. propose that these interactions allow synaptotagmin-1 to bend a section of the cell membrane in response to calcium ions.

The experiments also show that the C2B domain interacts more strongly with the SNARE complex than previously thought. A future challenge is to observe whether synaptotagmin-1 works in the same way in living cells.

Although the biochemical properties of Syt1 have been studied in detail, the functional importance of individual properties has remained unclear. It has been found that disrupting $Ca^{2+}$ binding to C2B impairs release much more strongly than disruption of C2A $Ca^{2+}$ binding sites in vivo (*Mackler et al., 2002*; *Nishiki and Augustine, 2004*; *Robinson et al., 2002*).Moreover, previous work showed that isolated C2B, instead of C2A, can promote SNARE-dependent membrane fusion in response to $Ca^{2+}$ in vitro (*Gaffaney et al., 2008*; *Xue et al., 2008*). These findings suggested that C2B plays a more preponderant role than C2A in neurotransmitter release. A number of studies have revisited this issue and suggested that the functional importance of C2B arises partly from its ability to deform membranes. For instance, C2B can induce vesicle clustering and/or membrane curvature in response to $Ca^{2+}$ (*Arac et al., 2006*; *Martens et al., 2007*; *Hui et al., 2009*; *Xue et al., 2008*). However, it is unclear why C2B has such membrane-deformation activity while C2A does not, given the fact that both C2 domains of Syt1 exhibit similar $Ca^{2+}$-dependent membrane-insertion properties in vitro.

Another potential reason for the striking functional asymmetry of the Syt1 C2 domains was provided by recent studies showing that interactions between C2B and SNARE complexes are crucial for the function of Syt1 in neurotransmitter release (*Zhou et al., 2015*; *Brewer et al., 2015*). However, these studies have yielded conflicting results. For instance, a recently solved Syt1–SNARE complex crystal structure showed a relatively large binding interface between Syt1 and the SNARE complex that involves two basic residues (Arg398 and Arg399, referred to as the R398 R399 region, see *Figure 1A*) on the bottom of C2B (*Zhou et al., 2015*), whereas another dynamic Syt1–SNARE complex structure model obtained by nuclear magnetic resonance (NMR) indicated a SNARE binding interface that is located on the polybasic patch (Lys326 and Lys327, referred to as the K326 K327 region, see *Figure 1A*) at the side of C2B (*Brewer et al., 2015*). Moreover, both of these basic regions have been previously implicated in binding to acidic membrane lipids, such as PS and PI(4,5) P2 (*Xue et al., 2008*; *van den Bogaart et al., 2011a*). Some other reports argued against the interaction between Syt1 and the SNARE complex, and suggested that the triggering function of Syt1 requires specific binding of C2B to acidic membranes rather than binding to SNARE complexes (*Honigmann et al., 2013*; *Park et al., 2015*). Taken together, although it seems clear that C2B can interact with either acidic SNARE complexes or acidic membranes due to the abundance of highly

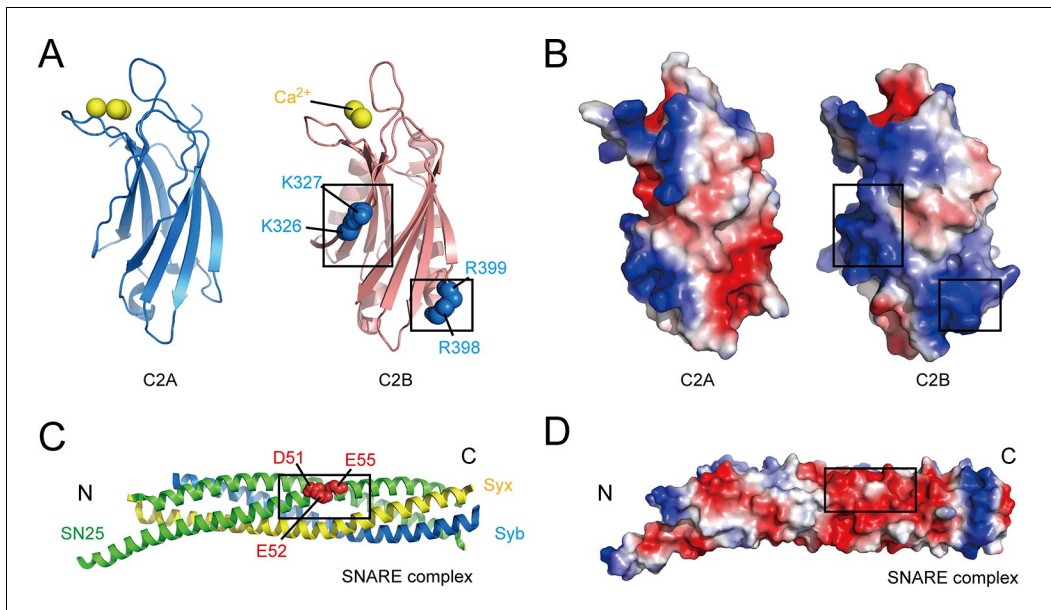

**Figure 1.** Overview of the structure features of Syt1 C2 domains and the core SNARE complex. (**A** and **B**) Structural diagrams (**A**) and electrostatic surface potential (**B**) of Syt1 C2A (PDB entry 1BYN) and C2B (PDB entry 1TJX) domain. Residues K326 and K327 on the side, R398 and R399 on the bottom, and $Ca^{2+}$ ions on the top of C2B are shown as blue and yellow spheres, respectively. Black boxes display the basic patches that include the residues shown in **A**. (**C** and **D**) Structural diagram (**C**) and electrostatic density map (**D**) of the core SNARE complex (PDB entry 1N7S). Residues D51, E52 and E55 are displayed as red spheres. Black box displays the acidic patch, which includes the residues shown in **C**. Syx, syntaxin-1; SN25, SNAP-25; Syb, synaptobrevin-2. The electrostatic surface potential was calculated by generating local protein contact potential (pymol software) and scaled from -5kT/e to 5kT/e, with red and blue denoting negative and positive potential, respectively.

positive charges around its surface (*Figure 1B*), the binding mode of C2B with SNARE complexes and membranes underlying the actual mechanism of Syt1 in release remains elusive.

To address these conundrums, we systematically investigated the binding properties of the C2B domain with SNARE complexes and membranes using diverse biophysical techniques. Our results showed that, prior to the $Ca^{2+}$ signal, the C2B domain interacts with PI(4,5)P2 and membrane-anchored SNARE complexes through the K326 K327 region and the R398 R399 region, respectively. We also found that the two interactions persist during insertion of the $Ca^{2+}$-binding loops into the membrane upon $Ca^{2+}$ influx. Consistent with a Syt1 triggering model proposed recently by Brunger and colleagues (*Zhou et al., 2015*), our data suggest that the local membrane deformation driven by the C2B domain constitutes a key step for triggering fusion.

## Results

### C2B interacts with SNARE complexes and acidic membranes in the absence of $Ca^{2+}$

As mentioned above, Syt1 C2B contains two highly basic regions at the side and the bottom of the structure (i.e., the K326 K327 and R398 R399 regions; see *Figure 1A,B*); both regions have been implicated in interactions with the SNARE complex (*Brewer et al., 2015*; *Zhou et al., 2015*). Conversely, an acidic patch located in the middle portion of the SNARE complex (e.g., residues D51, E52, and E55 on SNAP-25, likely with adjacent residues on syntaxin-1, see *Figure 1C,D*) has been suggested to mediate binding to C2B (*Brewer et al., 2015*; *Mohrmann et al., 2013*; *Zhou et al., 2015*). Using the GST pull-down assay, we first re-examined interactions between the two basic regions on C2B and the acidic patch on the assembled SNARE complex in the absence of $Ca^{2+}$. Consistent with previous results (*Brewer et al., 2015*), we found that both C2AB (soluble fragment of

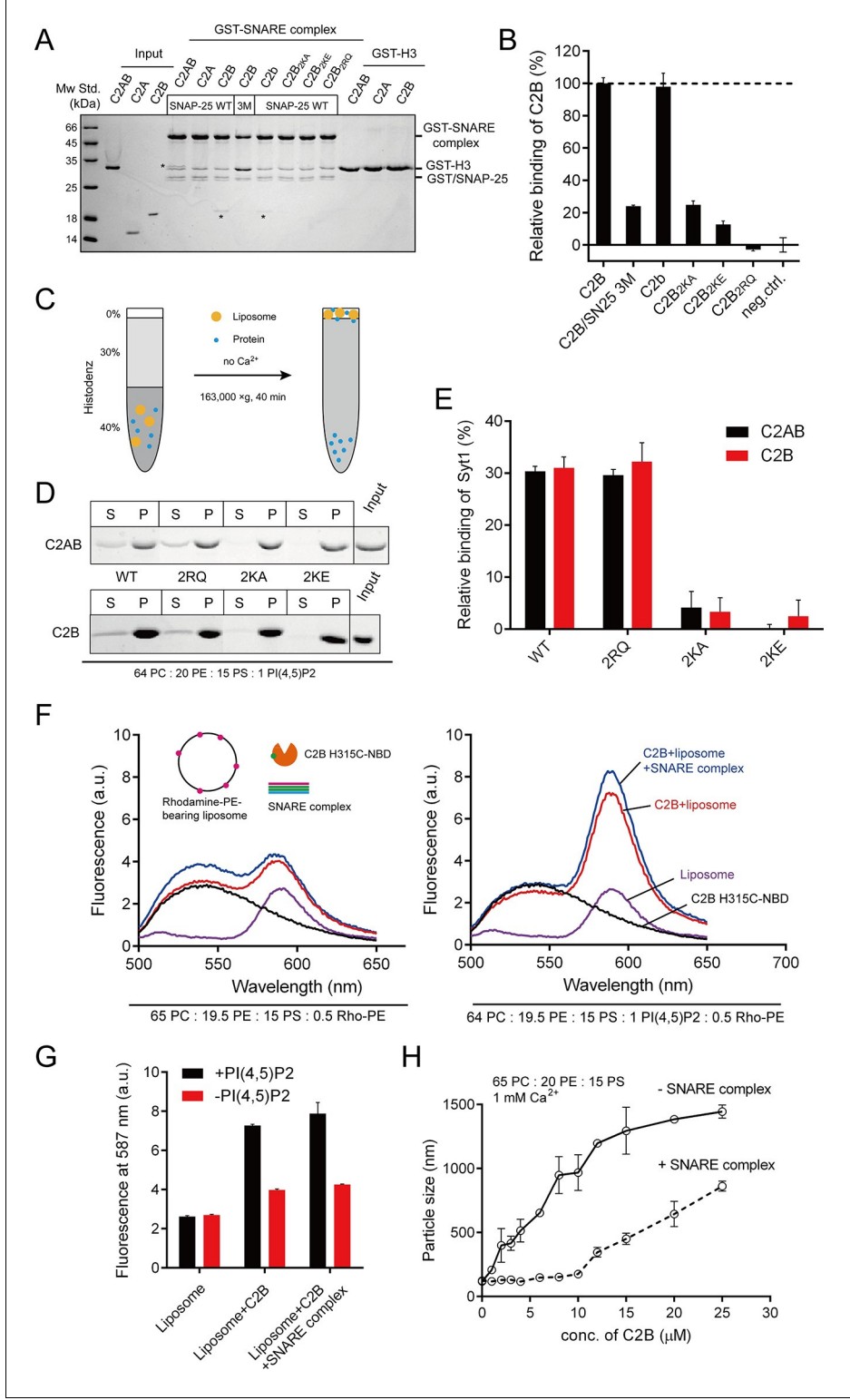

**Figure 2.** Different Ca²⁺-independent interactions of Syt1 with membranes and SNARE complexes. (**A** and **B**) Binding of Syt1 soluble fragments and their mutants to the core SNARE complex measured by GST pull-down assay (**A**) and quantification of the C2B binding (**B**). Asterisks in **A** show the bands of bound protein. 3M, GST-tagged SNARE complex bearing the SNAP-25 D51A/E52A/E55A mutation; H3, the SNARE motif of syntaxin-1; neg.ctrl., negative control, which represents C2B bound to GST-H3. Representative gel from one of three independent experiments is shown. Data are processed by Image J (NIH) and presented as the mean ± SD (n = 3),

*Figure 2 continued on next page*

*Figure 2 continued*

technical replicates. (C) Schematic diagram of the liposome co-flotation assay. After centrifuging, liposomes (orange) and bound proteins (blue) were co-floated on the top of the density gradients, remaining unbounded proteins left in the bottom of the gradients. (D and E) Co-flotation of C2AB, C2B and their mutants with liposomes bearing 1% PI(4,5)P2 in the absence of $Ca^{2+}$ (D) and quantification of the binding (E). WT/2RQ/2KA/2KE, Syt1 C2AB or C2B, and the mutants bearing R398Q/R399Q, K326A/K327A or K326E/K327E mutations, respectively; S, supernatant; P, pellet formed by centrifuging. Representative gel from one of three independent experiments is shown. Data are processed by Image J (NIH). (F and G) FRET between NBD labeled Syt1 C2B-H315C and rhodamine labeled liposomes with or without 1% PI(4,5)P2 (F) and quantification of the emission fluorescence of rhodamine at 587 nm (G). Liposome compositions are presented below the diagram; all reactions were performed in the absence of $Ca^{2+}$. (H) C2B-induced liposome clustering measured in the presence of SNARE complexes. The change in particle size as a function of the C2B concentration was measured by dynamic light scattering (DLS). All data plots are presented as the mean ± SD (n = 3), technical replicates.

The following figure supplements are available for figure 2:

**Figure supplement 1.** The SNAP-25 3M mutation displayed less resistance to SDS and no influence on SNARE complex assembly.

**Figure supplement 2.** PS-containing liposome clustering induced by C2B and its mutants in the presence of $Ca^{2+}$.

---

Syt1 harboring C2A and C2B) and C2B bound to assembled SNARE complexes, whereas C2A did not (*Figure 2A,B*, note that the C2A binding might be too weak to be detected on the gel). The SNARE complex containing the SNAP-25 D51A/E52A/E55A mutation (referred to as SN25 3M) showed impaired binding ability to C2B (*Figure 2A,B*), in agreement with previous findings (*Brewer et al., 2015*; *Mohrmann et al., 2013*; *Zhou et al., 2015*). Note that the SNARE complex bearing SN25 3M was less resistant to SDS (*Figure 2A* and *Figure 2—figure supplement 1A*), yet its assembly was normal (*Figure 2—figure supplement 1B,C*). As expected, disruption of the C2B $Ca^{2+}$-binding sites (D363N/D365N, referred to as C2b) caused no effect on the SNARE complex binding (*Figure 2A,B*). However, disruption of either the K326 K327 region or the R398 R399 region (K326A/K327A, K326E/K327E or R398Q/R399Q, referred to as C2B$_{2KA}$, C2B$_{2KE}$ or C2B$_{2RQ}$, respectively) impaired binding of C2B to the SNARE complex (*Figure 2A,B*), indicating that both basic regions of C2B contribute to the SNARE complex binding. This data reproduced previously contradictory findings (*Brewer et al., 2015*; *Zhou et al., 2015*), suggesting heterogeneous interactions between C2B and the SNARE complex in solution (*Park et al., 2015*).

This heterogeneous binding might be due to the absence of membrane lipids. PI(4,5)P2 selectively localizes to the plasma membrane, binds to Syt1 and plays crucial functions in neurotransmitter release (*van den Bogaart et al., 2011b*). Using a liposome co-flotation assay (*Figure 2C*), we found that, in the absence of $Ca^{2+}$, both C2AB and C2B (wild type, WT) bound efficiently to liposomes containing 1% PI(4,5)P2 (*Figure 2D,E*). This PI(4,5)P2-binding ability of Syt1 relies on the K326 K327 region but not the R398 R399 region, as C2AB$_{2KE}$/C2AB$_{2KA}$ or C2B$_{2KE}$/C2B$_{2KA}$ completely abolished binding to PI(4,5)P2-containing liposomes whereas C2AB$_{2RQ}$ or C2B$_{2RQ}$ did not (*Figure 2D,E*). To confirm this binding preference, we measured the fluorescence resonance energy transfer (FRET) between 7-nitrobenz-2-oxa-1,3-diazole (NBD) labeled C2B and rhodamine-labeled liposomes (*Hui et al., 2011*) containing 1% PI(4,5)P2 in the absence of $Ca^{2+}$. When NBD was placed close to the K326 K327 region (H315C-NBD), we detected a robust energy transfer (*Figure 2F,G*). Comparably, $Ca^{2+}$-independent FRET signals were considerably weaker when PI(4,5)P2 was removed from liposomes (*Figure 2F,G*). The addition of excess assembled SNARE complexes in the reaction caused no obvious effect on the FRET between C2B and PI(4,5)P2-containing liposomes (*Figure 2F,G*). These results, together with a recent finding that Syt1 binds to PI(4,5)P2-containing liposomes in physiological ionic conditions that contain ATP and $Mg^{2+}$ (*Park et al., 2015*), suggest that the K326 K327 region binds specifically to PI(4,5)P2-containing membranes in the absence of $Ca^{2+}$.

Given that the side K326 K327 region of C2B binds specifically to PI(4,5)P2 on membranes, we explored whether the bottom R398 R399 region tends to bind the SNARE complex. Previous studies suggested that the R398 R399 region binds acidic membranes (i.e., PS) and participates in liposome clustering (*Arac et al., 2006*; *Xue et al., 2008*). In our study using PS-containing liposomes, we

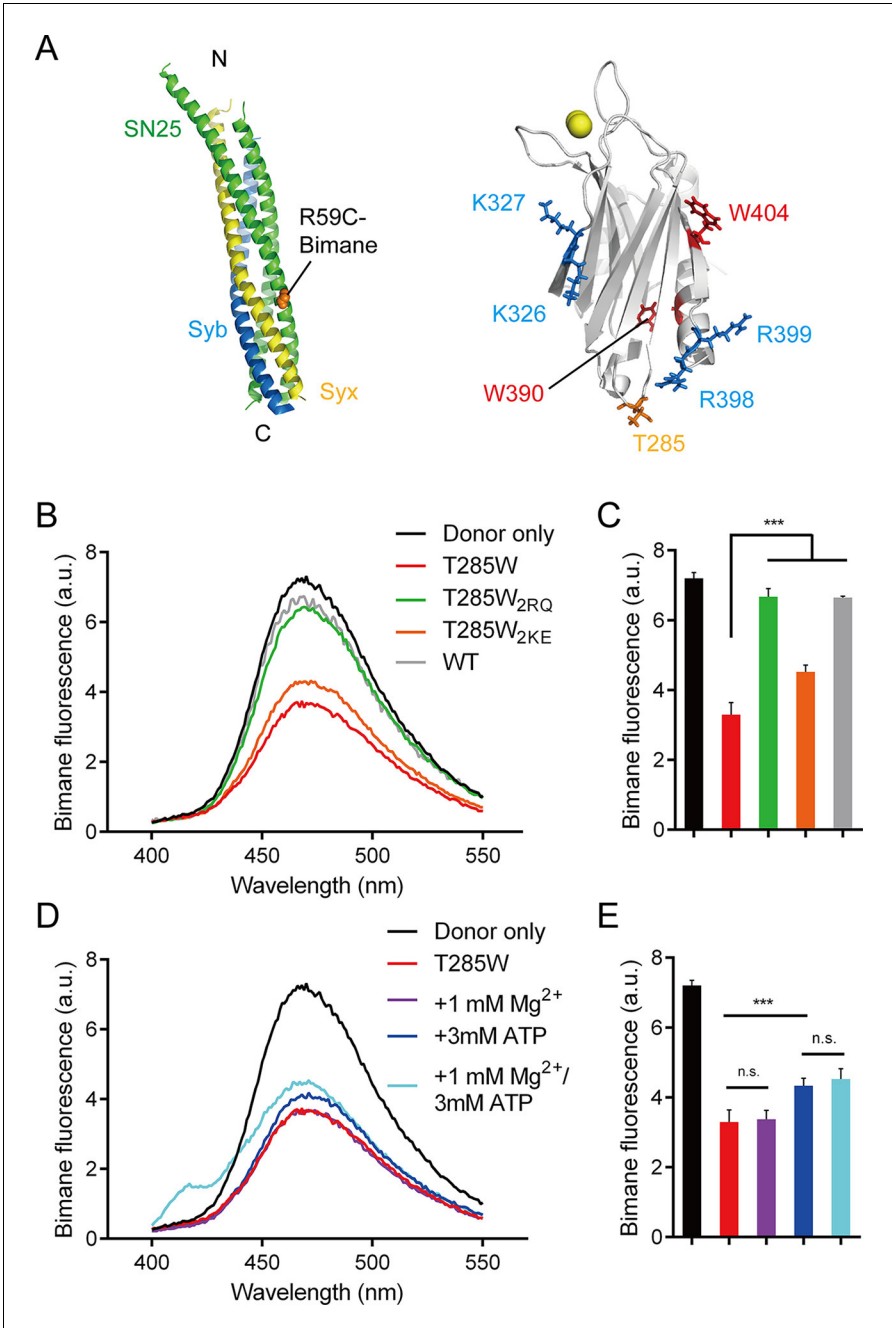

**Figure 3.** Persistence of the R398 R399–SNARE complex interaction in the presence of ATP and $Mg^{2+}$. (**A**) Schematic diagrams of bimane-labeled SNARE complex and Syt1 C2B. Tryptophan was introduced at the bottom of C2B (T285W, orange stick), which is close to residues R398 and R399; two native tryptophans (W390 and W404) are indicated as red sticks; residues K326 K327 and R398 R399 are shown as blue sticks; $Ca^{2+}$ ions are displayed as yellow spheres. (**B** and **C**) Quenching of bimane fluorescence on the SNARE complex with the addition of C2B T285W and the mutants in the absence of ATP and $Mg^{2+}$ (**B**) and quantification of the results (**C**). (**D** and **E**) Quenching of bimane fluorescence on the SNARE complex with the addition of C2B T285W in the presence of ATP and $Mg^{2+}$ (**D**) and quantification of the results (**E**). Donor only, no addition of Syt1 C2B; T285W, C2B bearing the T285W mutation; T285W$_{2RQ}$ and T285W$_{2KE}$, C2B T285W bearing the R398Q/R399Q or K326E/K327E mutations, respectively. Data are presented as the mean ± SD, technical replicates. n.s., not significant ($p > 0.05$); *$p<0.05$; ***$p < 0.001$; one-way ANOVA, n = 5.

indeed found that C2B clustered liposomes in response to $Ca^{2+}$ (**Figure 2—figure supplement 2**). Both $C2B_{2RQ}$ and C2b failed to cluster liposomes, while $C2B_{2KE}$ was able to cluster liposomes (**Figure 2—figure supplement 2**), supporting the idea that the bottom R398 R399 region and the top $Ca^{2+}$-binding loops associate two opposite acidic membranes in response to $Ca^{2+}$ (**Xue et al., 2008**). However, it is noteworthy that the PS-containing liposomes used in these experiments lacked the SNAREs (**Arac et al., 2006**; **Xue et al., 2008**). We therefore re-analyzed the liposome-clustering activity of C2B in the presence of soluble SNARE complexes, as a function of the concentrations of C2B. Robust clustering was observed as the C2B concentration increased in the presence of 1 mM $Ca^{2+}$ (**Figure 2H**). To our surprise, the addition of 10 µM soluble SNARE complexes in the reaction strongly impaired liposome clustering but the clustering was capable of gradually recovery as the concentration of C2B exceeded that of SNARE complexes (when C2B was above 10 µM, **Figure 2H**). Thus, the likely explanation is that the weaker R398 R399–PS interaction can be replaced by the stronger R398 R399–SNARE complex binding. These results suggest that the R398 R399 region binds preferentially to SNARE complexes rather than acidic membranes.

## Persistence of C2B (R398 R399)–SNARE complex interaction in the presence of ATP and $Mg^{2+}$

We further directly measured the C2B (R398 R399)–SNARE complex interaction using a bimane-tryptophan quenching assay in the absence of membranes. The bimane-tryptophan quenching assay has been previously used to study the structure and movements of proteins and has shown its sensitivity in short-distance electron transfer measurements (<10 Å) (**Islas and Zagotta, 2006**; **Mansoor et al., 2002**; **Taraska and Zagotta, 2010**). In this case, a single tryptophan mutation (T285W) at the bottom of C2B that is adjacent to the R398 R399 region was introduced (note that C2B contains two native tryptophans at residues 390 and 404 that are both far from the R398 R399 region, see **Figure 3A**). In addition, SNAP-25 was labeled with bimane via a single cysteine mutation (R59C) close to its acidic patch (D51 E52 E55), and was then assembled into the SNARE complex (**Figure 3A**). In contrast to a donor only condition (no addition of C2B) and the addtion of C2B (WT), the addition of C2B T285W induced robust quenching of bimane fluorescence on the SNARE complex (**Figure 3B,C**). Comparably, C2B T285W containing the R398Q/R399Q mutation (referred to as $T285W_{2RQ}$) showed strongly impaired binding to the SNARE complex, whereas C2B T285W containing the K326E/K327E mutation (referred to as $T285W_{2KE}$) did not have such an effect (**Figure 3B,C**). Consistent with previous results (**Zhou et al., 2015**), these data support the specific R398 R399–SNARE complex interaction.

We also investigated the R398 R399–SNARE complex interaction in the absence of membranes and in the presence of ATP and $Mg^{2+}$. Robust quenching of bimane fluorescence was still observed with the addition of C2B T285W (**Figure 3D,E**), suggesting that the specific R398 R399–SNARE complex interaction persists in a physiological ionic environment. Besides, the K326E/K327E mutation or ATP, but not $Mg^{2+}$, slightly influenced the interaction between the R398 R399 region and the SNARE complex (**Figure 3B–E**), likely owing to the electrostatic shielding. This data is inconsistent with a recent study showing that the Syt1–SNARE complex interaction is abolished at conditions containing ATP and $Mg^{2+}$ (**Park et al., 2015**). The reason for this contradiction is explained in the Discussion.

## Measurement of the binding affinity between C2B and SNARE complexes in the presence of membranes

Since C2B binds to PI(4,5)P2 and the SNARE complex in the absence of $Ca^{2+}$, we measured the binding affinity between C2B and the SNARE complex in the presence of membranes and in the absence of $Ca^{2+}$ by using the bimane-tryptophan quenching assay. Titration of C2B T285W to liposomes containing bimane-labeled *cis*-SNARE complexes yielded a $K_d$ of 1.53 ± 0.04 µM in the absence of PI(4,5)P2, and a $K_d$ of 0.86 ± 0.04 µM in the presence of PI(4,5)P2 (**Figure 4**), indicating a rather strong interaction between C2B (R398 R399) and the membrane-anchored *cis*-SNARE complex when PI(4,5)P2 is present. This higher binding affinity arises likely because PI(4,5)P2 helps recruit C2B to the membrane through the K326 K327 region and thereby increases the encounter between the R398 R399 region of C2B and the SNARE complex on membranes.

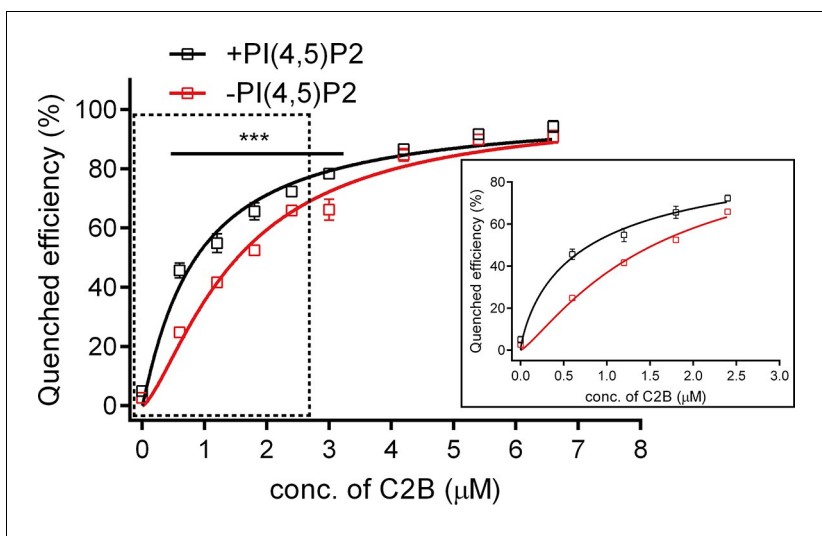

**Figure 4.** Binding $K_d$ between C2B and the membrane-anchored SNARE complex. *Cis*-SNARE complexes were reconstituted on liposome via the syntaxin-1 transmembrane domain. PI(4,5)P2 increased the binding affinity between Syt1 C2B and the membrane-anchored SNARE complex in the absence of $Ca^{2+}$. Plots show the quenched efficiency of the bimane-labeled *cis*-SNARE complex reconstituted on liposomes (65% PC + 20% PE + 15% PS) with the titration of Syt1 C2B T285W in the presence (black) and absence (red) of 1% PI(4,5)P2. Diagram in the solid box is the close-up view of the data in the dashed box. Plots are presented as the mean ± SD, technical replicates. \*\*\*p<0.001; multiple t-test using Holm-Sidak method, n = 5. Non-linear curve fit were achieved by the Michaelis-Menten equation where $V_{max}$ was constrained to 100 (% Quenched efficiency).

Taken together, all above binding results obtained in the absence of $Ca^{2+}$ support the idea that Syt1 'docks' two membranes via its transmembrane domain anchored on vesicles and its C2B domain binding to the plasma membrane (*Honigmann et al., 2013*; *van den Bogaart et al., 2011a*). It is conceivable that C2B pre-adsorbs to the plasma membrane prior to $Ca^{2+}$ influx, through its side K326 K327 region interacting with PI(4,5)P2, and its bottom R398 R399 region binding to surrounding SNAREs (e.g., SNAP-25) or assembled SNARE complexes.

## C2B interacts with SNARE complexes and acidic membranes in the presence of $Ca^{2+}$

We next sought to explore interactions of C2B with SNARE complexes and membranes in the presence of $Ca^{2+}$ using the liposome co-flotation assay. Previous studies have observed a strong binding of C2B to membranes owing to the insertion of the C2B $Ca^{2+}$-binding loops into PS-containing liposomes in the presence of 1 mM $Ca^{2+}$ (*Arac et al., 2006*; *Hui et al., 2011*). However, such strong binding would hinder the detection of interactions of C2B with PI(4,5)P2 or membrane-anchored SNARE complexes at the same time in our co-flotation experiments. To alleviate such an effect, we applied higher ion strength (250 mM KCl, instead of 150 mM KCl used in other experiments throughout the work) and weakened the $Ca^{2+}$-binding loops–$Ca^{2+}$–PS interaction by lowering the concentration of $Ca^{2+}$ from 1 mM to 0.1 mM (*Figure 5—figure supplement 1*). In the condition containing 250 mM KCl and 0.1 mM $Ca^{2+}$, we could not detect the interaction between C2B and PS-containing liposomes when the SNAREs were absent (*Figure 5A,B* and *Figure 5—figure supplement 1*). As such, we could not detect binding of C2B to membrane-anchored *cis*-SNARE complexes in the absence PS or $Ca^{2+}$ (*Figure 5A,B*). Efficient binding was only observed when *cis*-SNARE complexes and PS were both present on liposomes in the presence of 0.1 mM $Ca^{2+}$ (*Figure 5A,B*). Intriguingly, inclusion of 1% PI(4,5)P2 on liposomes that already contain PS and *cis*-SNARE complexes dramatically enhanced C2B binding in the presence of 0.1 mM $Ca^{2+}$, leaving very little C2B left in the pellet (*Figure 5C,D*). In contrast, selective removal of PS, PI(4,5)P2 or *cis*-SNARE complexes on liposomes led to strong impairment of C2B binding (*Figure 5C,D*), suggesting a synergy among the C2B ($Ca^{2+}$–binding loops)–$Ca^{2+}$–PS, C2B (K326 K327)–PI(4,5)P2 and C2B (R398 R399)–SNARE complex interactions in the presence of 0.1 mM $Ca^{2+}$.

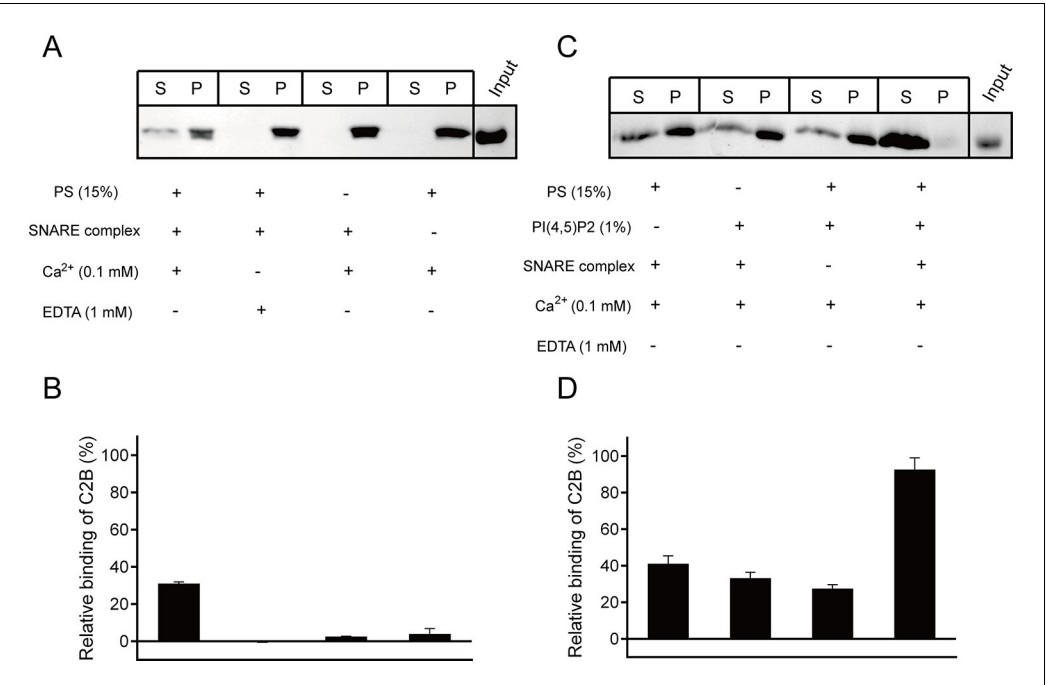

**Figure 5.** Synergistic interactions of C2B with membrane-anchored SNARE complexes, PI(4,5)P2 and PS in the presence of 0.1 mM $Ca^{2+}$. (A and B) Co-flotation of Syt1 C2B with liposomes in the absence of PI(4,5)P2 (A) and quantification of the results (B). (C and D) Co-flotation of Syt1 C2B with liposomes in the presence of 1% PI(4,5)P2 (C) and quantification of the results (D). *Cis*-SNARE complexes were reconstituted on liposome via the syntaxin-1 transmembrane domain. Liposomes compositions in A and C contain 65% PC, 20% PE, 15% PS with and without 1% PI(4,5)P2. S, supernatant; P, pellet. Representative gel from one of three independent experiments is shown. Data are processed by Image J (NIH) and presented as the mean ± SD (n = 3), technical replicates.

The following figure supplement is available for figure 5:

**Figure supplement 1.** Binding of C2B to PS-containing liposomes in different $Ca^{2+}$ concentrations.

## Persistence of both C2B–PI(4,5)P2 and C2B–SNARE complex interactions on membranes in response to $Ca^{2+}$

Next, we characterized the above synergistic interactions in the presence of 1 mM $Ca^{2+}$ and 150 mM KCl in a real-time manner. We first monitored insertion of the $Ca^{2+}$-binding loops into membranes in response to $Ca^{2+}$. For this purpose, we introduced cysteine mutations at the C2B T285W $Ca^{2+}$-binding loop 2 or 3 (T285W N333C or T285W I367C, respectively, see *Figure 6A*) and then labeled them separately with NBD and monitored their membrane-insertion abilities. Note that as an environment-sensitive probe, NBD exhibits a large increase in fluorescence intensity when it is transferred from an aqueous to a hydrophobic environment (*Crowley et al., 1993*). In the presence of 1 mM $Ca^{2+}$, addition of either T285W N333C-NBD or T285W I367C-NBD to liposomes containing PS and *cis*-SNARE complexes caused a marked increase in intensity (*Figure 6B,C*), confirming that the loops insert into membranes. In contrast, when additional 1% PI(4,5)P2 was applied, both T285W N333C-NBD and T285W I367C-NBD showed even higher intensities (*Figure 6D,E*), implying that the membrane-insertion ability of the $Ca^{2+}$-binding loops is enhanced with the assist of the C2B–PI(4,5)P2 interaction. Obviously, compared to T285W N333C-NBD, T285W I367C-NBD exhibited a remarkable enhancement in intensity in response to $Ca^{2+}$ (*Figure 6D,E*), suggesting that the loop 3 located on the same side as the K326 K327 region (see *Figure 6A*) on C2B is more accessible to membranes. These results imply that the K326 K327 region of C2B persistently sticks to PI(4,5)P2 on the plasma membrane with the insertion of the $Ca^{2+}$-binding loops into membranes. This supports the idea that $Ca^{2+}$-independent pre-adsorption of the K326 K327 region to PI(4,5)P2-harboring

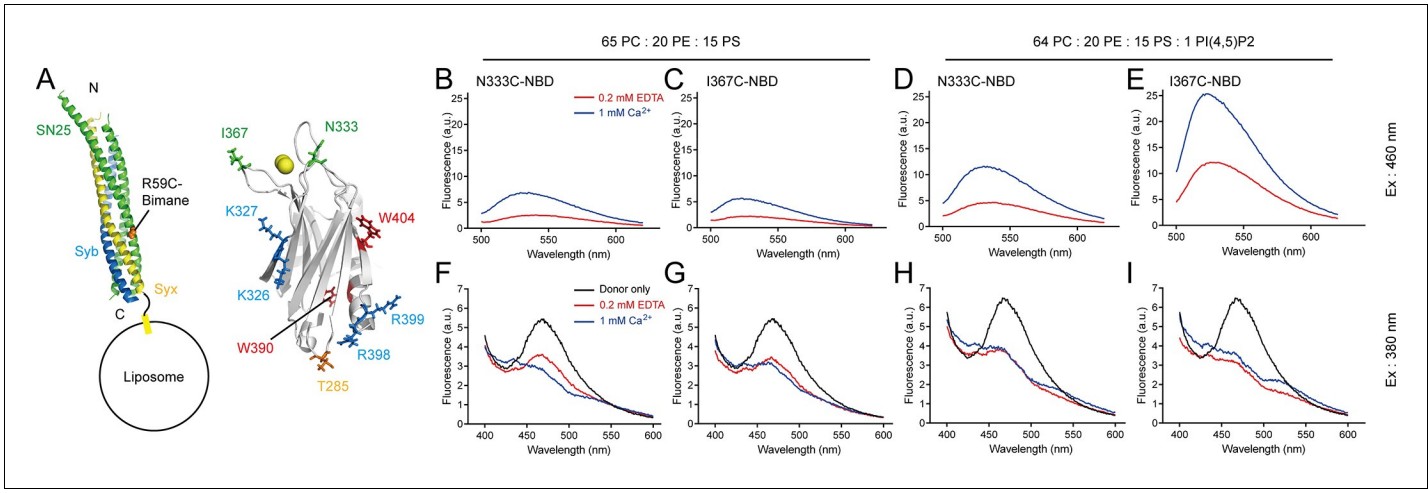

**Figure 6.** Persistence of C2B–SNARE complex and C2B–PI(4,5)P2 interactions upon insertion of the $Ca^{2+}$-binding loops into membranes. (**A**) Schematic diagrams of C2B and membrane-embedded *cis*-SNARE complex. NBD was labeled on N333C or I367C (green sticks) on C2B separately; bimane was labeled on SN25 R59C (orange sphere); tryptophan was introduced at the bottom of C2B (T285W, orange stick), which is close to residues R398 and R399; two native tryptophans (W390 and W404) are indicated as red sticks; residues K326 K327 and R398 R399 are shown as blue sticks; $Ca^{2+}$ ions are displayed as yellow spheres. (**B–E**) Detecting $Ca^{2+}$-triggered membrane insertion of the $Ca^{2+}$-binding loops using NBD fluorescence reporters in the absence (**B** and **C**) and presence (**D** and **E**) of PI(4,5)P2. Emission spectra were collected from 500 nm to 620 nm. (**F–I**) Detecting FRET between tryptophan (T285W) on C2B and bimane-labeled SNARE complexes reconstituted on liposomes in the absence (**F** and **G**) and presence (**H** and **I**) of PI(4,5)P2. Emission spectra were collected from 400 nm to 600 nm. Donor only, no addition of Syt1 C2B.

membranes helps 'steer' the $Ca^{2+}$-triggered membrane insertion of Syt1 toward the plasma membrane (*Bai et al., 2004*).

Meanwhile, we simultaneously monitored the interaction between C2B and membrane-anchored SNARE complexes using the bimane-tryptophan quenching assay when monitoring the C2B $Ca^{2+}$-binding loops inserting into membranes. The addition of T285W N333C-NBD or T285W I367C-NBD induced robust quenching of bimane fluorescence on membrane-anchored *cis*-SNARE complexes under all conditions with or without $Ca^{2+}$ and PI(4,5)P2 (*Figure 6F–I*), suggesting that the R398 R399 region binds consistently to the membrane-anchored SNARE complex before and after $Ca^{2+}$ influx. Indeed, crystal structures obtained in the absence and presence of $Ca^{2+}$ produced the same binding interface that involves residues R398 and R399 between Syt1 and the SNARE complex (*Zhou et al., 2015*). Together, these results suggest that the C2B (R398 R399)–SNARE complex interaction is $Ca^{2+}$ independent and this interaction persists during insertion of the $Ca^{2+}$-binding loops into the membrane.

In all, based on the findings that both the R398 R399–SNARE complex interaction and the K326 K327–PI(4,5)P2 interaction persist during entry of the $Ca^{2+}$-binding loops into membranes with PS, it is conceivable that the three simultaneous interactions of C2B induce membrane deformation in response to $Ca^{2+}$ (see the Discussion), as suggested by Brunger and colleagues (*Zhou et al., 2015*).

## $Ca^{2+}$-dependent simultaneous C2B–SNARE complex–membrane interactions are crucial for triggering fusion

The above results show that the C2B ($Ca^{2+}$–binding loops)–$Ca^{2+}$–PS, C2B (K326 K327)–PI(4,5)P2 and C2B (R398 R399)–SNARE complex interactions occur simultaneously in the presence of $Ca^{2+}$. We next sought to investigate whether all three interactions are required for the triggering activity of C2B in membrane fusion.

It has been widely reported that Syt1 C2AB or C2B can promote SNARE-dependent liposome fusion in response to $Ca^{2+}$ in vitro (*Chapman, 2008*; *Xue et al., 2008*; *Martens et al., 2007*). However, to what extent the promotion activity of C2AB (or C2B) in vitro reflects the triggering activity of Syt1 in vivo must be interpreted with caution because the liposome-clustering ability of C2AB (or C2B) correlated strongly with its activity in promoting SNARE-dependent liposome fusion (*Arac et al., 2006*; *Hui et al., 2011*; *Xue et al., 2008*). Accelerated liposome fusion might arise from

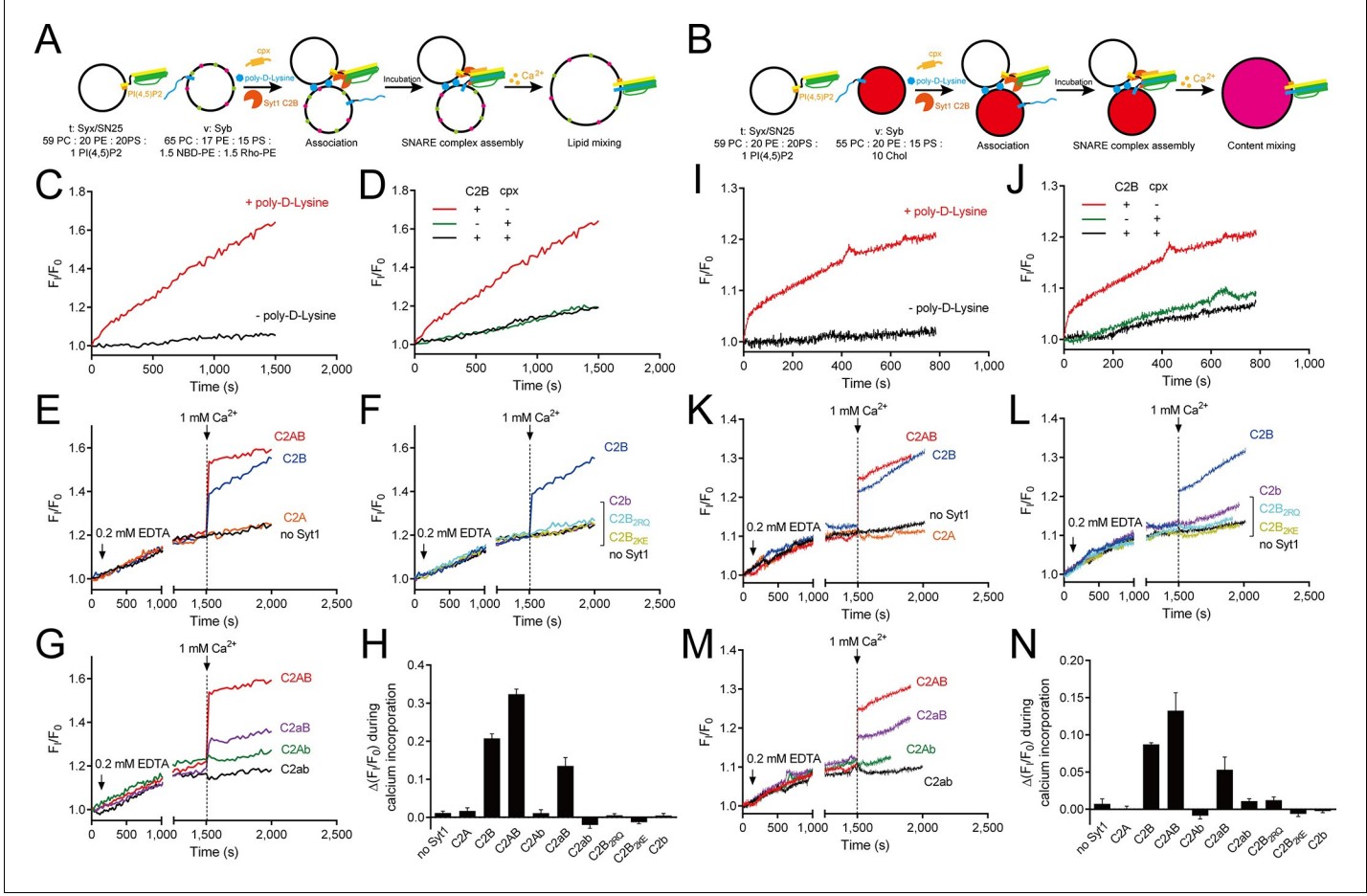

**Figure 7.** Ca$^{2+}$-dependent simultaneous C2B–SNARE complex–membrane interactions underlie the function of Syt1 in triggering fusion. (A and B) Schematic diagrams of the lipid mixing (A) and content mixing (B) assay. Liposome compositions are presented below the diagram. Cpx, complexin-1. (C and I) Poly-D-lysine promoted SNARE-dependent lipid mixing (C) and content mixing (I) in the absence of Ca$^{2+}$. (D and J) Cpx inhibited SNARE-dependent lipid mixing (D) and content mixing (J) in the absence of Ca$^{2+}$. (E and K) C2AB and C2B triggered fast lipid mixing (E) and content mixing (K) whereas C2A did not in response to Ca$^{2+}$. (F and L) Disruption of the C2B–SNARE complex–membrane interactions abolished fast lipid mixing (F) and content mixing (L). (G and M) The functional analysis of the Ca$^{2+}$-binding loops mutations on C2AB in triggering lipid mixing (G) and content mixing (M). (H and N) Quantification of the lipid-mixing (H) and content-mixing results (N) in E–G and K–M, respectively. Data are presented as the mean ± SD (n = 3), technical replicates.

The following figure supplements are available for figure 7:

**Figure supplement 1.** Liposome clustering induced by Poly-D-lysine in a concentration-dependent manner.

**Figure supplement 2.** Liposome clustering and SNARE pairing monitored during liposome fusion.

**Figure supplement 3.** No leakiness of liposomes detected in the content-mixing experiments.

the enhanced membrane docking and SNARE pairing caused by C2AB (or C2B) (*Hui et al., 2011*). To detect the actual activity of C2AB (or C2B) in triggering fusion, in our liposome fusion system (*Figure 7A,B*) we utilized poly-D-lysine as a Ca$^{2+}$-independent factor to mimic liposome docking and increase SNARE pairing (*Hui et al., 2011*); added complexin-1 to arrest liposomes in a 'ready-for-fusion' state (*Kyoung et al., 2011*; *Diao et al., 2012*; *Lai et al., 2014*); and finally applied 1 mM Ca$^{2+}$ to trigger fusion. A lower concentration of C2AB or its mutants (0.5 µM) was applied in the system to avoid liposome clustering.

The fusion between liposomes (t-liposomes) reconstituted with syntaxin-1–SNAP-25 complex and synaptobrevin-containing liposomes (v-liposomes) was monitored (*Figure 7A,B*) by using a FRET

based lipid-mixing and content-mixing assay (*Ma et al., 2013*; *Yang et al., 2015*). Consistent with previous observations (*Hui et al., 2011*), addition of 2 µg/ml poly-D-lysine in the mixtures efficiently induced liposome clustering (*Figure 7—figure supplement 1*), thereby enhancing SNARE-dependent lipid mixing and content mixing in the absence of $Ca^{2+}$ (in the presence of 0.2 mM EDTA, *Figure 7C,I*). This data is consistent with the idea that liposome clustering can promote fusion through facilitating SNARE pairing. Complexin-1 inhibited both lipid mixing and content mixing in the presence of poly-D-lysine and in the absence of $Ca^{2+}$ (*Figure 7D,J*), whereas further addition of 1 mM $Ca^{2+}$ at 1500 s triggered both lipid mixing and content mixing (*Figure 7E,K*). Consistent with previous studies (*Gaffaney et al., 2008*; *Xue et al., 2008*), the triggering activity strictly required C2B instead of C2A (*Figure 7E,K*). We emphasize that C2B was unable to cluster SNARE-bearing liposomes at a concentration of 0.5 µM in this fusion system (*Figure 7—figure supplement 2A*), and *trans*-SNARE complexes were mostly assembled at the ready-for-fusion stage arrested by complexin-1 (*Figure 7—figure supplement 2B*). In addition, SNARE pairing was not obviously promoted during the triggering (*Figure 7—figure supplement 2B*). These results pinpoint the triggering activity of C2B in membrane fusion. Furthermore, to exclude the possibility that content-mixing signals represented by de-quenching of sulforhodamine fluorescence arise from liposome leakiness, we performed control experiments where both v- and t-liposome were loaded with sulforhodamine (*Figure 7—figure supplement 3A*). Leakiness was not detected in SNARE-dependent content mixing in the presence of poly-D-lysine with and without C2AB–$Ca^{2+}$ (*Figure 7—figure supplement 3B,C*), indicating that the real membrane fusion events were observed in our experiments.

Using this system, we investigated whether the simultaneous interactions of C2B with PS, PI(4,5) P2, and the SNARE complex are critical for the triggering activity of C2B. We found that the $Ca^{2+}$-binding sites mutation (C2b) strongly impaired the ability of C2B to trigger both lipid mixing and content mixing in response to $Ca^{2+}$ (*Figure 7F,L*). Similar results were obtained when using C2B$_{2KE}$ or C2B$_{2RQ}$ mutations (*Figure 7F,L*). These results are consistent with those physiological data in previous studies (*Li et al., 2006*; *Xue et al., 2008*; *Zhou et al., 2015*), suggesting that the simultaneous interactions of C2B with the SNARE complex and membranes are crucial for the triggering function of Syt1.

In addition, we assessed the functional importance of the $Ca^{2+}$-binding loops of C2A *versus* C2B in triggering liposome fusion with this system. We found that disrupting the C2A $Ca^{2+}$-binding sites (D230N/D232N, C2aB) caused a moderate impairment in the lipid mixing and content mixing, whereas disrupting the C2B $Ca^{2+}$-binding sites (D363N/D365N, C2Ab) totally abolished the ability of C2AB to trigger fusion (*Figure 7G,M*). Disruption of the $Ca^{2+}$-binding sites on both C2 domains (D230N/D232N/D363N/D365N, C2ab) completely abolished the triggering effect (*Figure 7G,M*). Quantification of the lipid-mixing and content-mixing activities of Syt1 and its mutants are shown in *Figure 7H, N*. Thus, this fusion system successfully reconstituted the triggering role of C2AB and C2B in vitro and reproduced the relative importance of the C2 domain $Ca^{2+}$-binding sites observed in vivo (*Mackler et al., 2002*; *Nishiki and Augustine, 2004*; *Robinson et al., 2002*; *Shin et al., 2009*).

Actually, previous in vitro studies (*Kyoung et al., 2011*; *Diao et al., 2012*; *Zhou et al., 2015*) have successfully reconstituted the full-length Syt1–SNARE complex machinery at physiological Syt1 copy number and also included complexin-1. To complement these studies, we used the fusion system described in *Figure 7A,B* with C2AB replaced by full-length Syt1 (reconstituted on v-liposomes) (*Figure 8A,B*). Besides, poly-D-lysine was excluded because full-length Syt1 can 'dock' two membranes (*van den Bogaart et al., 2011a*). We found that Syt1$_{AB}$ (WT) and its mutations on the $Ca^{2+}$-binding sites (Syt1$_{aB}$, Syt1$_{Ab}$ and Syt1$_{ab}$, respectively) all stimulated lipid mixing and content mixing in the absence of $Ca^{2+}$ and complexin-1 (*Figure 8C,F*), consistent with the docking role of Syt1. As expected, complexin-1 arrested liposomes bearing Syt1 at the ready-for-fusion stage, and further addition of 1 mM $Ca^{2+}$ at 1500 s triggered fusion (*Figure 8D,G*). Consistent with the results observed in our C2AB-based fusion experiments (*Figure 7*), Syt1$_{AB}$ and Syt1$_{aB}$ efficiently triggered lipid mixing and content mixing while Syt1$_{Ab}$ and Syt1$_{ab}$ did not (*Figure 8D,E,G,H*), verifying the functional importance of the C2B $Ca^{2+}$-binding sites in $Ca^{2+}$-triggered liposome fusion. Note that a physiological ratio of full-length Syt1 and the SNAREs were reconstituted in our experiments (*Figure 8I*) as previously reported (*Lai et al., 2014*; *Zhou et al., 2015*).

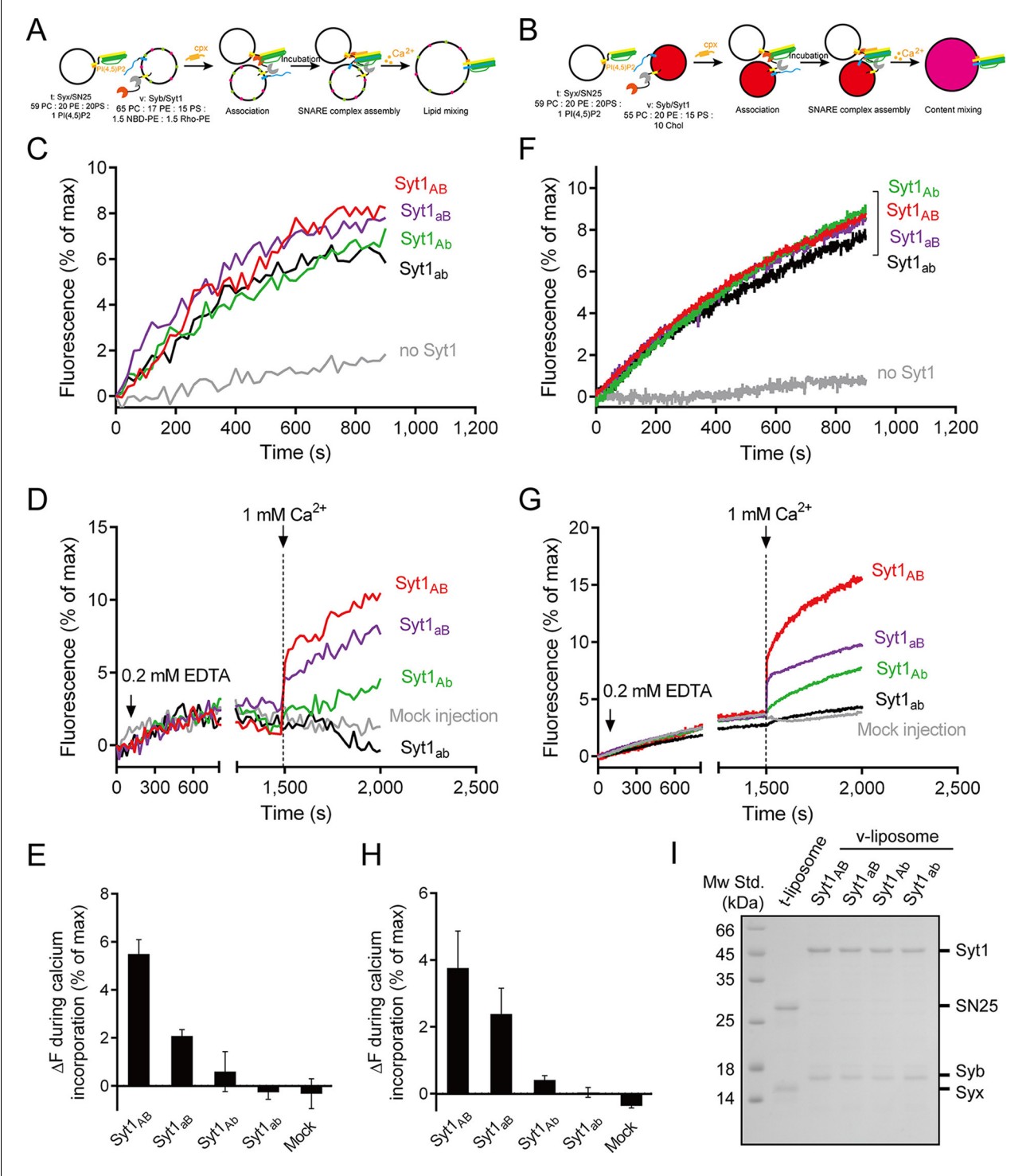

**Figure 8.** Functional analysis of the Ca²⁺-binding loops on full-length Syt1 in triggering liposome fusion. (**A** and **B**) Schematic diagrams of the lipid mixing (**A**) and content mixing (**B**). (**C** and **F**) Syt1 stimulates lipid mixing (**C**) and content mixing (**F**) in the absence of $Ca^{2+}$ and complexin-1 (cpx). (**D** and **G**) The functional analysis of the $Ca^{2+}$-binding loops on Syt1 full-length in triggering lipid mixing (**D**) and content mixing (**G**). (**E** and **H**) Quantification of the lipid-mixing (**E**) and content-mixing results (**H**) in **D** and **G**, respectively. Data are presented as the mean ± SD (n = 3), technical replicates. (**I**) Analysis of reconstituted proteins on liposomes by SDS-PAGE. Mock injection represents the addition of the buffer (no $Ca^{2+}$) instead of $CaCl_2$ during triggering.

## Discussion

Syt1 acts as a $Ca^{2+}$ sensor and plays key functions in neurotransmitter release through its C2 domains. The C2 domains exhibit similar overall structures and $Ca^{2+}$-induced membrane-insertion properties but differ strikingly in their function during release. Despite the fact that the functional significance of the C2B domain in vivo has been successfully reproduced in SNARE-dependent membrane fusion assays in vitro the mechanism by which C2B acts with the SNAREs and membranes to promote fusion is unclear. In the present study, we suggest a membrane-bending property of the C2B domain that arises from its simultaneous interactions with SNARE complexes and membranes. The relevance of our proposed C2B–SNARE complex–membrane interactions is supported by the present study and many previously reported data (*Radhakrishnan et al., 2009*; *van den Bogaart et al., 2011a*; *Zhou et al., 2015*), concordantly with the increasing realization that Syt1 cooperates with the SNARE complex and membranes in neurotransmitter release.

The functional importance of C2B arises most likely from its unique structure that contains abundant basic residues on its surface, which endows C2B with the ability to bind acidic SNARE complexes and/or membranes. Contiguously electrostatic potentials created by the bottom R398 R399 residues and the side K326 K327 residues of C2B contribute to the SNARE binding, as observed in *Figure 2A,B*. However, as demonstrated with our co-flotation and FRET experiments (*Figure 2C–G*), the presence of PI(4,5)P2-containing membranes shifts the equilibrium towards an energetically favorable binding where the K326 K327–PI(4,5)P2 interaction dominates. The much higher efficiency of the K326 K327 region, compared to the R398 R399 region, in PI(4,5)P2 binding arises most likely because the K326 K327 region contains a much higher positive-charge density, which enables it to bind tightly to lipid head groups with a highly negative-charge density [i.e., PI(4,5)P2-microdomains at active zones] (*Honigmann et al., 2013*; *Joung et al., 2012*; *Park et al., 2012*). Consistent with this, further investigations on the C2B–SNARE complex interaction using a more sensitive bimane-tryptophan quenching assay (*Figure 3* and *4*) indicate that the R398 R399 region of C2B binds to the SNARE complex or the membrane-anchored SNARE complex in a $Ca^{2+}$-independent manner. Thus, the existence of the K326 K327–PI(4,5)P2 interaction and the R398 R399–SNARE complex interaction prior to $Ca^{2+}$ influx likely recruits Syt1 to the fusion sites, which underlies the docking function of Syt1, as suggested previously (*de Wit et al., 2009*; *Honigmann et al., 2013*).

A recent report (*Park et al., 2015*) argued against the Syt1–SNARE complex interaction because this interaction measured in the study appeared to be completely abolished in the presence of ATP and $Mg^{2+}$. However, Syt1 used in the study was labeled at residue 342, which is close to the polybasic patch, suggesting that this study actually measured the FRET between the K326 K327 region and the SNARE complex, and it is unlikely that it reflects the real Syt1–SNARE complex interaction. Instead, our study found that the R398 R399–SNARE complex interaction persists in the absence and presence of ATP and $Mg^{2+}$ (*Figure 3*). It is also noteworthy that C2B T285W and SNAP-25 R59C mutations used in our bimane-tryptophan quenching assay seem unlikely to affect the particular interaction between Syt1 and the SNARE complex, because these residues are outside the 'primary' interface (interface area: 720 $Å^2$; including residues Arg398 and Arg399) between Syt1 and the SNARE complex (*Zhou et al., 2015*). In addition, the $C^{\alpha}$–$C^{\alpha}$ distance between the two labeling sites is measured at ∼12 Å based on the Syt1–SNARE complex structures (*Zhou et al., 2015*), consistent with the relatively large effect on the FRET observed in our experiments. Moreover, we measured a reasonably strong binding $K_d$ [0.86 ± 0.04 μM and 1.53 ± 0.04 μM in the presence and absence of PI(4,5)P2, respectively] between C2B and the SNARE complex in the presence of membranes (*Figure 4*). Thus, our binding results, together with the observations that both spontaneous and $Ca^{2+}$-evoked release are not affected by the presence of 3 mM ATP (*Zhou et al., 2015*), strongly suggest that the interaction between Syt1 and the SNARE complex is not affected by ionic shielding and is physiologically relevant.

The finding that the R398 R399 region binds preferentially to the SNARE complex in this study seems to be incompatible with our previous studies (*Arac et al., 2006*; *Xue et al., 2008*). This discrepancy may arise from the different experimental conditions used between the present work and our previous studies. Re-examination of liposome clustering in a more stringent condition containing abundant SNARE complexes showed that the liposome-clustering ability of C2B is totally abrogated (*Figure 2H*). This data suggests that the R398 R399–PS binding might be displaced by the R398 R399–SNARE complex interaction. However, our results could not completely rule out the possibility

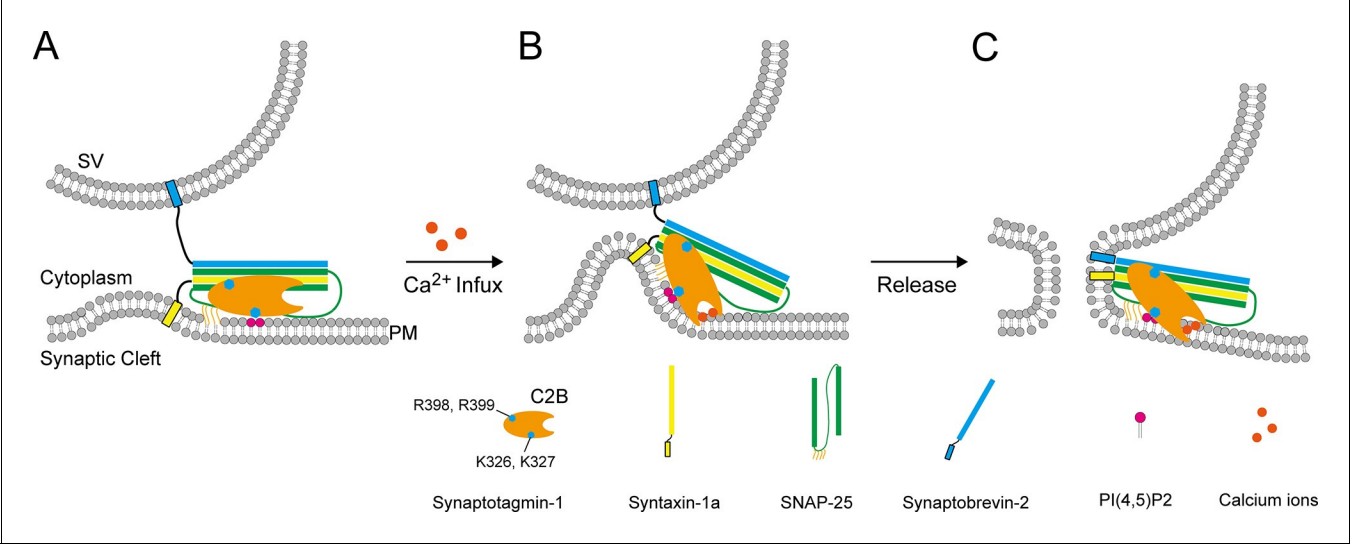

**Figure 9.** A working model of Syt1 in triggering membrane fusion. (**A**) Binding of Syt1 C2B to PI(4,5)P2 and primed *trans*-SNARE complex on the plasma membrane before Ca²⁺ influx. (**B**) The simultaneous interactions of C2B with primed *trans*-SNARE complex and PI(4,5)P2-PS-containing membranes in response to Ca²⁺ cause bucking toward the synaptic vesicle of the plasma membrane. Note that a similar model has been recently proposed (*Zhou et al., 2015*). (**C**) Membrane bucking might cooperate with the action of Syt1 C2B in displacing inhibitory complexin-1 to facilitate the continuous helical SNARE complex assembly, thus triggering membrane fusion and neurotransmitter release. SV, synaptic vesicle; PM, pre-synaptic membrane.

that a small population of Syt1 molcules act to shorten the distance between membranes via the direct interaction of the R398 R399 region with acidic phospholipids in response to Ca²⁺.

By detecting penetration of C2B into membranes (with PS) as well as binding of C2B to the SNARE complex and PI(4,5)P2 at the same time (*Figures 5* and *6*), our results provides acceptable proof for the persistence of the K326 K327–PI(4,5)P2 interaction and the R398 R399–SNARE complex interaction during Ca²⁺ influx. The presence of the simultaneous SNARE-containing membrane binding of the top Ca²⁺-binding loops, of the side K326 K327 region, and of the bottom R398 R399 region, leads to a possible membrane-deformation mechanism of Syt1 (*Figure 9*): before Ca²⁺ influx, both the K326 K327–PI(4,5)P2 interaction and the R398 R399–SNARE complex interaction fasten C2B in an orientation parallel with the plasma membrane, leaving the Ca²⁺-binding loops held back from membrane insertion (*Figure 9A*); Ca²⁺ influx strongly induces fast insertion of the Ca²⁺-binding loops into membranes, rearranging C2B to an orientation vertical to the plasma membrane. The persistent binding of C2B to PI(4,5)P2 and the membrane-anchored SNARE complex would thus exert a membrane bending force to buck the local membrane outward in response to Ca²⁺ (*Figure 9B*). Furthermore, it is likely that binding of C2B to PI(4,5)P2 and SNARE complexes before Ca²⁺ influx would lead to a ring-like arrangement of C2B molecules around the fusion pore (*Wang et al., 2014*), which would then facilitate bucking local membranes upon collective bending forces in response to Ca²⁺. This possible membrane-deformation mechanism is in good agreement with the Syt1 working model proposed by Brunger and colleagues (*Zhou et al., 2015*), and is supported by a recent observation that local membrane protrusions (5 nm in height, similar to the size of one C2B molecule) bucked on the surface of GUV (giant unilamellar vesicles) require the presence of Syt1 and assembled SNARE complexes (*Bharat et al., 2014*).

The proposed membrane-deformation mechanism is supported by the finding that all three interactions of C2B are required for the triggering function of the Syt1 in liposome fusion (*Figure 7*). This membrane-deformation mechanism explains very well the recent observation that isolated C2B readily bends membranes (*Hui et al., 2009*), and suggests that membrane insertion of the Ca²⁺-binding loops alone is not sufficient to drive membrane deformation without the assist of the R398 R399 and the K326 K327 regions. Thus, the functional importance of the C2B Ca²⁺-binding sites observed in previous in vivo studies and our present liposome fusion experiments (*Figures 7* and *8*) can be explained: disruption of the C2B Ca²⁺-binding sites abrogates the simultaneous interaction of C2B

with acidic membrane lipids [i.e., PI(4,5)P2 and PS] and the membrane-anchored SNARE complex, so that the ability of C2B to bend membranes is absolutely abolished. Our data reinforce the idea that local membrane deformation by $Ca^{2+}$–Syt1 is key for the triggering function of Syt1 in release (*Hui et al., 2009*; *Zhou et al., 2015*). Our data also reinforce the notion that the coordinated efforts of two or more interactions from one protein can induce membrane deformation (*McMahon and Boucrot, 2015*).

The liposome fusion results support the notion that SNARE complexes are already partially assembled before $Ca^{2+}$ influx (*Figure 7* and *Figure 7—figure supplement 2B*), which enables complexin-1 binding and thereby strains such a complex in a stage ready for fusion (*Rizo and Xu, 2015*). In response to $Ca^{2+}$, C2B-induced membrane bucking would reduce the distance between two apposed membranes, which might cooperate with the action of C2B in displacing inhibitory complexin-1, to facilitate the continuous helical SNARE complex assembly that propagates through the linker region into the transmembrane domains (*Figure 9C*). Thus, high curvature stresses induced by C2B and the energy released from the C-terminal SNARE complex assembly might be coupled together in response to $Ca^{2+}$ to overcome the energy barrier for membrane fusion. It is noteworthy that C2A would have an ancillary role by binding to one membrane and helping to dictate the apparent $Ca^{2+}$ affinity of Syt1 (*Robinson et al., 2002*; *Zhou et al., 2015*). Altogether, our results add increasing evidence for the triggering mechanism by which Syt1 acts in concert with the SNARE complex and membranes to promote membrane fusion.

## Materials and methods

### Recombinant protein purification

The cytoplasmic domain of rat Syt1 (known as C2AB) used in this study comprises residues 140–421, and the C2A and C2B domains comprise residues 140–266 and 270–421, respectively. All Syt1 fragments or their mutants, full length rat synaptobrevin-2 and its cytoplasmic domain (residues 29–93), the H3 domain of rat syntaxin-1a (residues 191–253) and full-length rat complexin-1 were constructed into pGEX-6p-1 vector (GE Healthcare; Piscataway, NJ); full-length rat Syt1 (all cysteines were mutated to alanine except the cysteine residue at position 277), rat C-terminal syntaxin-1a (residues 183–288, without Habc domain), full-length human SNAP-25a (with its four cysteins mutated to serines) and SNAP-25a 3M (D51A/E52A/E55A) were constructed into pET28a vector (Novagen; Australia); rat syntaxin-1a C-terminal (residues 183–288, without Habc domain) and human SNAP-25a (with its four cysteins mutated to serines) or SNAP-25a 3M were constructed into pETDuet-1 vector (Novagen). All the recombinant proteins above were expressed in *E.coli* BL21 DE3 cells and purified as previously described (*Lai et al., 2014*; *Ma et al., 2013*; *van den Bogaart et al., 2011a*). Point mutations were prepared by using the QuickChange Site-Directed Mutagenesis Kit (Agilent Technologies; Santa Clara, CA).

### GST pull-down assay

Purified GST-H3 (residues 191–253 of syntaxin-1a) was incubated with SNAP-25 or SNAP-25 3M (SN25 3M) and synaptobrevin (residues 29–93) overnight and analyzed by SDS-PAGE to confirm the SNARE complex formation. 20 µM GST-SNARE complex or GST-H3 was mixed with 10 µM Syt1 fragments and 20 µl 50% (v/v) Glutathione Sepharose 4B affinity media (GE Healthcare) to a final volume of 50 µl. After 2 hr gentle shaking at 4°C, beads were washed 3 times using 25 mM HEPES pH 7.4, 150 mM KCl, and 10% glycerol (buffer A). Samples were analyzed by SDS-PAGE. All experiments were performed in the absence of $Ca^{2+}$.

### Liposomes preparation

Lipid powder (all from Avanti Polar Lipids; Alabaster, AL) was dissolved in chloroform at a concentration of 10 mg/ml for storage at -20°C, except for brain PI(4,5)P2 (from porcine's brain) in chloroform: methanol:water 20:9:1 at 1 mg/ml. Lipids were mixed at the proper ratio as indicated in the figures or legends to a final concentration of 5 mM and dried under nitrogen followed by vacuum for at least 3 hr. Lipid films were dissolved in buffer A containing 0.2 mM Tris (2-carboxyethyl) phosphine (TCEP, Sigma Aldrich; St. Louis, MO) and 1% CHAPS (w/v, Amresco; Solon, OH) and vortexed for 5 min. For preparing proteoliposomes, purified proteins dissolved in 1% CHAPS (w/v) were added

into the dissolved lipid films to a final protein-to-lipid ratio of 1:200 (for SNAREs) and/or 1:1000 (for Syt1 full-length), respectively; for plain liposomes, equivalent buffer A containing 1% CHAPS (w/v) was added; after 30 min incubation at room temperature, the mixtures were dialyzed against buffer A containing 0.1 mM TCEP and 1.0 g/L Bio-beads (Bio-Rad; Hercules, CA) at 4°C 3 times. The prepared proteoliposomes were checked using Dynamic Light Scattering (DLS) on a DynaPro Nanostar (Waytt Technology, Santa Barbara, CA) before using.

## Liposome co-flotation assay

Liposome (2 mM total lipids) compositions are indicated in the figures or legends. Liposomes were incubated with 10 µM proteins (±$Ca^{2+}$) in buffer A (unless stated otherwise) for 40 min at room temperature. The liposomes and bound proteins were isolated by flotation on a Histodenz (Sigma Aldrich) density gradients (40%:30%) using a SW 55 Ti rotor (Beckman Coulter; Boulevard Brea, CA) at 163,000 ×g for 40 min. Samples from the top and the bottom of the gradient (20 µl) were taken and analyzed by SDS-PAGE and Coomassie Brilliant Blue (CBB) staining.

## Fluorescence measurements

For liposome-protein FRET experiments (*Figure 2F,G*), 100 µM liposomes were mixed with 5 µM Syt1 C2B H315C-NBD and 20 µM soluble SNARE complex. Fluorescence was monitored in a physiological ion condition (buffer A) on a PTI QM-40 fluorescence spectrophotometer (PTI; Edison, NJ) with an excitation wavelength of 460 nm and an emission spectra from 500 to 650 nm.

For bimane-tryptophan electron transfer in the absence of membranes (*Figure 3*), 1 µM assembled SNARE complex which harbors monobromobimane (mBBr, Molecular Probes; Eugene, OR) labeled SNAP-25a R59C was mixed with 2 µM Syt1 C2B or its mutant (as indicated in the figures), and additional 1 mM magnesium chloride (analytical grade) and 3 mM ATP (Bio Basic Inc.; Canada) were incorporated as indicated. Fluorescence was monitored on a PTI QM-40 fluorescence spectrophotometer with an excitation wavelength of 380 nm and emission spectra from 400 to 600 nm.

For the $K_d$ measurement between Syt1 C2B and membrane-anchored SNARE complex in *Figure 4*, Syt1 C2B T285W was mixed with 200 µM liposome [64% POPC, 20% POPE, 15% DOPS and/or 1% PI(4,5)P2, removed PI(4,5)P2 was supplied with POPC] bearing bimane-labeled SNARE complex (with a protein-to-lipid ratio of 1:200) with indicated concentration. Bimane fluorescence was monitored on a PTI QM-40 fluorescence spectrophotometer with an excitation wavelength of 380 nm and an emission wavelength of 470 nm. Data plots were fitted using the Michaelis-Menten equation, where $V_{max}$ was constrained to 100 (% Quenched efficiency).

For bimane-tryptophan electron transfer and NBD membrane-insertion assay (*Figure 6*), 100 µM liposomes (bearing bimane-labeled SNARE complex with syntaxin-1 transmembrane domain anchored on liposomes and with a protein-to-lipid ratio of 1:200) were mixed with 5 µM NBD-labeled Syt1 C2B. A dual excitation of 380 nm (for bimane) and 460 nm (for NBD) and a dual emission spectrum of 400–600 nm and 500–620 nm was used to collect the fluorescence of bimane and NBD, respectively.

Fluorescence anisotropy assay in *Figure 2—figure supplement 1* was carried out as previously described (*Wiederhold and Fasshauer, 2009*). 200 nM BODIPY FL (Molecular Probes) labeled synaptobrevin (residues 29–93, S61C) was mixed with 1 µM syntaxin-1 (residues 191–253) and SNAP-25 or 1 µM pre-incubated syntaxin-1–SNAP-25 complex.

For the SNARE-pairing assay shown in *Figure 7—figure supplement 2B*, 0.2 mM EDTA, 2 µg/ml poly-D-lysine, 20 µM complexin and 0.5 µM Syt1 C2B were incorporated into a mixture of t-liposome (100 µM lipids and 0.5 µM syntaxin-1 S200C-tetramethylrhodamine [TMR, Molecular Probes]-SNAP-25) and v-liposome (50 µM lipids and 0.25 µM synaptobrevin D44C-BODIPY FL) unless otherwise indicated. After incubation for 1480 s, 1 mM $Ca^{2+}$ was added. Donor fluorescence was monitored with an excitation wavelength of 485 nm and an emission wavelength of 513 nm. All lipid compositions are indicated in the figures or legends. All experiments were performed at 25°C in a 1-cm quartz cuvette in buffer A.

## Liposome clustering assay

Liposome clustering assay was carried out as previously described (*Arac et al., 2006*; *Xue et al., 2008*). Briefly, 100 µM liposomes were mixed with 1 mM $Ca^{2+}$, with/without 10 µM of a soluble

SNARE complex or the equivalent volume of buffer and the indicated concentration of Syt1 C2B was incorporated for 40 min incubation at room temperature. Liposome compositions are indicated in the figures or legends. Particle sizes were analyzed by DLS using a DynaPro Nanostar (Waytt Technology) at 25°C.

### Lipid mixing assay

General procedures are indicated in *Figure 7A* and *Figure 8A*. For lipid mixing using the soluble Syt1 fragments, 0.2 mM EDTA, 2 ug/ml poly-D-lysine, 20 μM complexin and 0.5 μM Syt1 fragments were added to a mixture of 100 μM t-liposomes (bearing 0.5 μM syntaxin-1–SNAP-25) and 50 μM v-liposomes (bearing 0.25 μM synaptobrevin). For lipid mixing using reconstituted full-length Syt1, poly-D-lysine was excluded, and v-liposomes (50 μM) were reconstituted with synaptobrevin (bearing 0.25 μM synaptobrevin) and 0.05 μM full-length Syt1 (with a protein-to-lipid ratio of 1:1000). After incubation, 1 mM $Ca^{2+}$ was added to trigger lipid mixing at 1500 s. Donor (NBD) fluorescence were monitored on a PTI QM-40 fluorescence spectrophotometer with an excitation wavelength of 460 nm and an emission wavelength of 538 nm. Fluorescence in *Figure 7* was normalized to the initial fluorescence intensity. Fluorescence in *Figure 8* were normalized to the fluorescence intensity at 0.1% Triton X-100. All experiments were carried out at 25°C in buffer A. Lipid compositions are indicated in the figures or legends.

### Content mixing assay

General procedures are indicated in *Figure 7B* and *Figure 8B*. 40 mM sulforhodamine B (Sigma) was loaded into v-liposome (harboring synaptobrevin with or without full-length Syt1) without lipid probes. Other details are the same as with lipid mixing assays. Leakiness control was performed with 40 mM sulforhodamine B both loaded into t-liposomes and v-liposomes. Fluorescence was monitored on a PTI QM-40 fluorescence spectrophotometer with an excitation wavelength of 565 nm and an emission wavelength of 580 nm. Fluorescence normalization is the same as that used in the lipid-mixing assay. All experiments were carried out at 25°C in buffer A. Lipid compositions are indicated in the figures or legends. Additional 10% Cholesterol (Chol, from ovine wool, Avanti Polar Lipids) was incorporated into v-liposomes to prevent leakiness.

### Statistical analysis

Prism 6.01 (Graphpad) and Image J (NIH) were used for graphing and statistical tests, all of which are described in figure legends.

## Acknowledgements

We thank Josep Rizo (University of Texas Southwestern Medical Center at Dallas) for providing constructs and insightful advice. We thank Li Yu and Senfang Sui (Tsinghua University) for helpful discussion.

## Additional information

### Funding

| Funder | Grant reference number | Author |
| --- | --- | --- |
| National Science Foundation of China | 31322034 | Cong Ma |
| National Key Basic Research Program of China | 2014CB910203 | Cong Ma |
| National Key Basic Research Program of China | 2015CB910800 | Cong Ma |

The funders had no role in study design, data collection and interpretation, or the decision to submit the work for publication.

## Author contributions

SW, Conception and design, Acquisition of data, Analysis and interpretation of data; YL, Acquisition of data, Analysis and interpretation of data; CM, Conception and design, Analysis and interpretation of data, Drafting or revising the article

## Author ORCIDs

Shen Wang, http://orcid.org/0000-0002-5013-1039
Cong Ma, http://orcid.org/0000-0002-7814-0500

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
