## [Decision Letter]

[Editors’ note: a previous version of this study was rejected after peer review, but the authors submitted for reconsideration. The first decision letter after peer review is shown below.]

Thank you for submitting your work entitled "Synaptotagmin C2B domain cooperates with the SNARE complex and membranes to promote fusion" for consideration by *eLife*. Your article has been reviewed by three peer reviewers, one of whom is a member of our Board of Reviewing Editors, and the evaluation has been overseen by Richard Aldrich as the Senior Editor. Our decision has been reached after consultation between the reviewers. Based on these discussions and the individual reviews below, we regret to inform you that your work in its present form is not suitable for publication in *eLife*.

There are two topics mixed in this paper: (1) simultaneous/cooperative binding of Syt1-C2B to both SNARE complex and membrane, and (2) enhancement of SNARE complex formation by Munc13 conjunction with Syt1. This work would greatly benefit from splitting these new results into two papers. The reviewers and editors felt that the results on Syt1-C2B-membrane interactions are potentially more interesting than the results on Munc13. Several concerns (as outlined below) were raised by all reviewers which preclude further consideration of the manuscript in its present form.

Reviewer #1:

This work presents data that suggest that the C2B domain of Syt-1 simultaneously and cooperatively interacts with both the SNARE complex and the plasma membrane prior to Ca^2+^ influx. The results and discussions in this manuscript may clarify previous results in the literature that suggested competition between membrane binding and SNARE complex binding. This work also suggests that Syt1 C2B stimulates Munc18-Munc13-dependent SNARE complex assembly in the absence of calcium.

Comments:

A better discussion of the previous findings by Zhou et al. (2015) should be included in the Introduction and the Discussion: Zhou et al. found that there is a conserved and calcium independent interface between synaptotagmin-1 and the neuronal SNARE complex and that this interface is essential for fast synchronous release in neurons. Moreover, Zhou et al. proposed that the C2B-SNARE complex unit exists prior to calcium binding and then moves as a block, deforming the membrane by simultaneous three interactions with the membrane involving the polybasic region of Syt1, the calcium binding loops of Syt1 C2B, and the membrane anchor and juxtamembrane region of syntaxin and SNAP-25, similar to the model proposed in this manuscript.

In the model Figure 7, the orientation of the C2B domain is inconsistent with the crystal structure of the primary interface by Zhou et al. (2015) (the C2B domain is rotated by 180 degrees compared to the model shown in Figure 7). Realizing that the data presented in this manuscript are insufficient to determine the orientation of the C2B domain and the authors probably wrote this paper before the Zhou et al. publication became available, the model should nevertheless be corrected according to the crystal structure in order to avoid confusion.

Figure 5 shows that the C2B – SNARE interaction is Ca^2+^ independent. This actually has been shown by Zhou et al. since crystal structures in the absence and presence of Ca^2+^ produced the same primary interface between Syt1 and SNARE comple (please cite this in this context); What is new here is that the interaction persists upon insertion of the Ca binding loops into the membrane.

The results presented in this paper are in conflict with the Brewer et al. (NSMB 2015) results. The authors are encouraged to discuss possible reasons for this apparent conflict.

The lipid composition of the co-flotation assay (Figure 2) apparently did not include PS, whereas the other experiments (2F, 2H) include 15% PS. Is there a reason that PS was omitted in the experiment shown in Figure 2? Ideally, PS should be included in all experiments for consistency.

A recent paper (Park Y, et al. (2015), Nat Struct Mol Biol. doi: 10.1038/nsmb.3097) implies that Syt1 – SNARE interactions are weak or non-existent at 150 mM KCl, MgCl2 and ATP (Figure 5 in the Park et al. paper). What are the specific salt condition for the various Syt1-SNARE binding experiments in Figure 2 and Figure 5? Ideally, the conditions of the Park et al. paper should be tested in the experiments described in Figure 5.

In Figure panels 3C, D, experiments with the C2B 2KE and 2RQ mutants should be performed, similar to what was done in panels 3E, F.

In Figure 5, controls should be performed using the K326E, K327E (polybasic – membrane binding) and R398Q R399Q (SNARE binding) mutants.

Figure 5: what is the lipid composition of the proteoliposomes?

The bulk assay shown in Figure 6 may be subject to effects caused by vesicle docking and/or lipid mixing since both effects would result in the observed decrease in donor fluorescence. Content mixing assays should be performed.

Reviewer #2:

The role of patches of basic residues at the side (K326, K327) and bottom (R398, R399) of the synaptotagmin 1 C2B domain on SNARE protein/complex and phospholipid binding have been focus of numerous studies in the recent past, leading to partially conflicting results. This study aims to resolve the SNARE and membrane binding modes of the C2B domain (in absence and presence of Ca^2+^) and how these interactions contribute to the fusion process. The authors utilize a number of techniques GST pulldown, coflotation, fluorescence measurements and in vitro fusion assays to demonstrate that mutation of side and bottom basic residues impair SNARE complex binding and membrane binding in a Ca dependent manner.

This work is based on a large body of literature and the findings presented here are in part confirmatory and in part new.

New is the finding that the interactions of the bottom loop with the membrane can be competitively displaced by the presence of SNARE complexes. Another aspect of the work is that the authors show that synaptotagmin C2B domain is able to double the rate of SNARE complex formation. Perhaps the most interesting aspect is that they can simultaneously monitor both penetration of C2B domain into the membrane and binding to the SNARE complex with the bottom loop at the same time, providing decent proof for the presence of a simultaneous binding of top loops to membranes and bottom loop to cis-SNARE complex via the acid patch created by SNAP25 residues. However, does this experiment exclude that a considerable fraction of the bottom loop would also be able to interact with the phospholipid membrane?

On other issue here is the conversion of positive to negative charges in the mutations: net charge difference particularly in the coflotation assays may be of concern.

*Reviewer #3:*

This study has important new data, is largely well-written and, in certain respects, presents major advances, especially in Figure 3.

1) They "propose that the membrane bending induced by the fast transition between the two binding modes underlies the [fusion] triggering function of C2B" (from the Abstract), yet there is no measurement of membrane bending nor is there any data which implicates the transition between the two binding modes, as opposed to the binding modes themselves, in causing fusion. They are welcome to make such a model, of course, but should make clearer distinctions between what they show and what is simply hypothesis.

2) The data in Figure 6 from the classical assay of dequenching between two fluorescent lipids purports to show "lipid mixing". However, it has been shown in this journal (Zick and Wickner, *eLife* 2014;3:e09251) that this assay measures trans-SNARE pair dependent events which do not correspond to lipid mixing. In the text, they assume that this lipid mixing is the result of fusion, though it was shown in this same paper that such dequenching can occur in the absence of fusion. The authors should replace this figure with one measuring the mixing of proteoliposomal lumenal contents (in the continuous presence of an external quenching agent, to control for leakage or lysis).

3) They claim that calcium binding "…causes a cooperative interaction of C2B with membranes and the SNARE complex." The data however show that C2B can bind membranes or SNARE complex independently, with no data showing cooperativity to this binding.

4) Figure 2; they claim that C2B but not C2A binds to SNARE complex, but the C2B band is faint and the C2A band could be just below the detection limit. Also, they claim synergy between the 2 basic regions of C2B for their binding to SNARE complex, but there is no quantitative binding data to gird this claim. What do they mean by synergy here?

5) Does "MUN", as used in Figure 3, refer to both Munc13 and Munc18? Were these singly omitted, to support the claim that both are needed? I don't see this in the figure.

[Editors’ note: what now follows is the decision letter after the authors submitted for further consideration.]

Thank you for submitting your work entitled "Synaptotagmin C2B domain interacts simultaneously with SNAREs and membranes to promote fusion" for consideration by *eLife*. Your article has been reviewed by three peer reviewers, one of whom is a member of our Board of Reviewing Editors, and the evaluation has been overseen by Richard Aldrich as the Senior Editor.

The reviewers have discussed the reviews with one another and the Reviewing Editor has drafted this decision to help you prepare a revised submission.

Summary:

This is a re-submission of a previous submission to *eLife*. The authors have now focused on Syt1 – SNARE – membrane interactions, making this a more accessible and interesting study for the general reader. The authors revisited three proposed key interactions of the Syt-1 C2B domain – (i) calcium-dependent binding of the calcium binding loops to membranes with PS, (ii) binding of basic residues K326/327 to membranes with PIP2, and (iii) binding of basic residues R398/399 to the SNARE complex. The data indicate that all three interactions occur simultaneously and are all required for the triggering function the Syt-1 C2B domain in liposome/vesicle fusion. The findings reconcile recently published contradictory data on the functional significance of SNARE complex binding by Syt-1 (e.g. Park et al., 2015 vs. Zhou et al., 2015) – the discrepancy of the present data with those of Park et al. (2015) as regards the ion-sensitivity of the Syt-1-SNARE interaction is explained by differences in the areas labeled for fluorescence/FRET-based interaction studies.

In addition, the authors present new data on the *K_d_* between Syt1 and the SNARE complex in the presence of membrane. To our knowledge this is the first time a reasonably strong *K_d_* has been reported for this interaction (below 1 microM), compared to the relatively weak Syt1-SNARE interaction in the absence of membranes. However, there are a few issues that need to be addressed before acceptance.

Essential revisions:

1) The pull-down assay shown in Figure 2 is used as a basis to claim that mutation of the acidic patch in the SNARE complex (SNAP-25 D51/E52/E55 to A – 3M) abolishes binding to the Syt-1 C2B domain. However, it looks as if there is much less SNARE complex in this assay. The authors claim that this is due to lower SDS-resistance of the 3M complex, but there is no trace of the SNARE monomers on the gel, and there is a (slight) difference in the assembly between the WT and the 3M complex. How can one be sure that the same amount of assembled SNARE complex was added in the assay?

2) Figure 3: In their binding assays designed to show the synergistic effects of the three key Syt-1 C2B binding modes, the authors used a condition where the dominant effect of the calcium-dependent interaction is reduced. How sensitive are the interactions to calcium (see also next point)?

3) Figure 3 implies that there is no interaction between C2B and the membrane-reconstituted SNARE complex in the absence of calcium and absence of PIP2. Yet, Figure 4 suggests that there is such an interaction in the absence of calcium and presence of PIP2. The authors should attempt a co-flotation experiment using the same conditions as that of Figure 4, and in the absence of calcium.

4) Figure 4: Syt1-T285W and SNAP-25 R59C mutations. The authors should comment in the text that these residues are outside the primary interface between Syt1 and SNARE complex as observed in the crystal structures by Zhou et al. (2015). Thus, these mutations are unlikely going to affect this particular interaction between Syt1 and SNARE complex. This is an important comment since unfortunate placement of labels have affected this interaction in work by others (Brewer et al., 2015; Park et al., 2015). Moreover, the Ca-Ca distance between the two labeling sites is ~ 12 A (based on the structures by Zhou et al.), consistent with the relatively large effect on FRET that the authors observe. Again, this is worth mentioning.

5) Figure 6: Was this experiment done in the presence of membrane (i.e., similar to the method shown in Figure 4)?

6) Figure 7 and second paragraph subsection “Ca^2+^-dependently simultaneous C2B–SNARE complex–membrane interactions are crucial for triggering fusion”: The in vitro assay by Kyoung et al., 2011; Diao et al., 2012; Lai et al., 2014) actually reconstituted the full-length Syt1-SNARE machinery at physiological Syt1 copy number and also included complexin. Moreover, Zhou et al. (2015) showed that mutation of the primary interface disrupts calcium-triggered vesicle fusion in this assay. These results should be mentioned in the context of presenting Figure 7 and the authors should point out which aspects of their experiments go beyond the previous results by Zhou et al. (2015) (e.g., the studies of calcium-binding site mutations.

7) Figure 7: Based on other work (Wang Z, Liu H, Gu Y, Chapman ER (2011) Reconstituted synaptotagmin I mediates vesicle docking, priming, and fusion. J Cell Biol 195:1159-70. doi: 10.1083/jcb.201104079) the behaviors of the isolated Syt-1 C2B or Syt-1 C2AC2B fragments are different from the behavior of full-length Syt-1. The authors need to validate the findings in Figure 7 that go beyond those of Zhou et al. (2015) by using full-length, reconstituted Syt-1.

8) Figure 8: the palmitoylation positions of the SNAP-25 linker region are incorrectly drawn – they are close to the C-terminal side of the SNARE complex.

9) Please provide the rationale for the (relative) concentrations of SNAREs, SNARE complexes, and Syt-1 fragments used in the different assays.

[Editors' note: further revisions were requested prior to acceptance, as described below.]

Thank you for resubmitting your work entitled "Synaptotagmin^-1^ C2B domain interacts simultaneously with SNAREs and membranes to promote membrane fusion" for further consideration at *eLife*. Your revised article has been favorably evaluated by Richard Aldrich (Senior editor), and two reviewers, one of whom is a member of our Board of Reviewing Editors. The manuscript has been improved but there are some small remaining issues that need to be addressed before acceptance, as outlined below:

1) Please move the new results with full-length synaptotagmin^-1^ to the main figure and move the results with the soluble C2AB domain (Figure 7) to a supplement. The results with the full-length protein are more relevant.

2) Results, paragraph one: "K326E/K327E, K326E/K327E" should be "K326A/K327A, K326E/K327E".

---

## [Author Response]

[Editors’ note: the author responses to the first round of peer review follow.]

There are two topics mixed in this paper: (1) simultaneous/cooperative binding of Syt1-C2B to both SNARE complex and membrane, and (2) enhancement of SNARE complex formation by Munc13 conjunction with Syt1. This work would greatly benefit from splitting these new results into two papers. The reviewers and editors felt that the results on Syt1-C2B-membrane interactions are potentially more interesting than the results on Munc13. Several concerns (as outlined below) were raised by all reviewers which preclude further consideration of the manuscript in its present form.

In the new manuscript, we took out Munc13 results and only focused on the results on C2B–SNARE complex–membrane interactions as suggested by the editors. In addition, we have tried to address most of the concerns the reviewers raised and we have included data from new experiments that were performed based on some of the reviewers’ comments. Below I summarize the revisions made and provide answers to other reviewer concerns.

Reviewer #1:

*This work presents data that suggest that the C2B domain of Syt-1 simultaneously and cooperatively interacts with both the SNARE complex and the plasma membrane prior to Ca^2+^ influx. The results and discussions in this manuscript may clarify previous results in the literature that suggested competition between membrane binding and SNARE complex binding. This work also suggests that Syt1 C2B stimulates Munc18-Munc13-dependent SNARE complex assembly in the absence of calcium. Comments: A better discussion of the previous findings by Zhou et al. (2015) should be included in the Introduction and the Discussion: Zhou et al. found that there is a conserved and calcium independent interface between synaptotagmin^-1^ and the neuronal SNARE complex and that this interface is essential for fast synchronous release in neurons. Moreover, Zhou et al. proposed that the C2B-SNARE complex unit exists prior to calcium binding and then moves as a block, deforming the membrane by simultaneous three interactions with the membrane involving the polybasic region of Syt1, the calcium binding loops of Syt1 C2B, and the membrane anchor and juxtamembrane region of syntaxin and SNAP-25, similar to the model proposed in this manuscript.*

We included the finding by Zhou et al. (2015) in the Introduction and mentioned the model proposed by Zhou et al. (2015) throughout the paper.

*In the model Figure 7, the orientation of the C2B domain is inconsistent with the crystal structure of the primary interface by Zhou et al. (2015) (the C2B domain is rotated by 180 degrees compared to the model shown in Figure 7). Realizing that the data presented in this manuscript are insufficient to determine the orientation of the C2B domain and the authors probably wrote this paper before the Zhou et al. publication became available, the model should nevertheless be corrected according to the crystal structure in order to avoid confusion.*

We redrew the Syt1 triggering model (now in Figure 8) according with the C2B–SNARE complex primary binding interface provided by the Syt1–SNARE complex structure (Zhou., et al. (2015)). The orientation of the C2B domain is thus accordingly corrected by a 180-degrees rotation with respect to the SNARE complex, as suggested by the model proposed by Zhou., et al. (2015).

*Figure 5 shows that the C2B – SNARE interaction is Ca^2+^ independent. This actually has been shown by Zhou et al. since crystal structures in the absence and presence of Ca^2+^ produced the same primary interface between Syt1 and SNARE comple (please cite this in this context); What is new here is that the interaction persists upon insertion of the Ca binding loops into the membrane.*

We thank the reviewer for reminding us and we cited this paper in the according place.

*The results presented in this paper are in conflict with the Brewer* et al. *(NSMB 2015) results. The authors are encouraged to discuss possible reasons for this apparent conflict.*

As shown in our present study, both the bottom R398 R399 region and the side K326 K327 region of C2B are able to interact with the SNARE complex in solution (Figure 2), as reported in the literature. Thus, we suppose that interactions between C2B and the SNARE complex in solution are dynamic unless the appearance of acidic membranes. We discussed this issue in the Discussion paragraph two.

*The lipid composition of the co-flotation assay (Figure 2) apparently did not include PS, whereas the other experiments (2F, 2H) include 15% PS. Is there a reason that PS was omitted in the experiment shown in Figure 2? Ideally, PS should be included in all experiments for consistency.*

As suggested, we re-tested the binding of C2AB/C2B and the mutants with liposomes containing both PIP2 and PS with the co-flotation assay and confirmed that the K326 K327 region mediates the PIP2 binding (Figure 2).

*A recent paper (Park Y, et al. (2015), Nat Struct Mol Biol. doi: 10.1038/nsmb.3097) implies that Syt1 – SNARE interactions are weak or non-existent at 150 mM KCl, MgCl2 and ATP (Figure 5 in the Park et al. paper). What are the specific salt condition for the various Syt1-SNARE binding experiments in Figure 2 and Figure 5? Ideally, the conditions of the Park et al. paper should be tested in the experiments described in Figure 5.*

First, the salt conditions for the Syt1–SNARE binding experiments were 25 mM HEPES pH 7.4, 150 mM KCl, 10% glycerol in Figure 2 and Figure 4–Figure 6, and was 25 mM HEPES pH 7.4, 250 mM KCl, 10% glycerol in Figure 3. Second, a recent report (Park Y, et al. (2015)) argues against the physiologically relevant of the Syt1–SNARE complex interaction by showing that this binding is abolished in conditions containing ATP and Mg^2+^ with FRET experiments. However, C2AB used in the study was labeled at residue 342, which is close to the polybasic region, so that this study measured the interaction between the K326 K327 region and the SNARE complex. These data only suggest the competition of ATP/Mg^2+^ with the SNARE complex for the K326 K327 region binding. In addition, Park Y, et al. suggested in their study that the K326 K327 region prefers to PIP2 binding at a physiologically ionic environment. Our data support the notion that K326 K327 region binds to PIP2 (Figure 2) and showed that the interaction between the R398 R399 region and the SNARE complex persisted even at conditions containing 25 mM HEPES pH 7.4, 150 mM KCl, 10% glycerol, 3mM ATP and 1mM Mg^2+^ (Figure 6), reinforcing the functional importance of the Syt1–SNARE complex interaction.

*In Figure 5, controls should be performed using the K326E, K327E (polybasic – membrane binding) and R398Q R399Q (SNARE binding) mutants.*

In Figure 5 (now in Figure 4), instead of using the K326E, K327E mutant, we used liposomes that lacked PIP2 as the control because these liposomes lost the ability to bind the K327 K327 region. In addition, we accordingly tested the interaction between the R398 R399 region and the SNARE complex in solution by using the K326E/K327E and R398Q/R399Q mutations (Figure 6).

*Figure 5: what is the lipid composition of the proteoliposomes?*

The lipid composition of the proteoliposomes are shown on the top of the panels in Figure 5 (now in Figure 4).

*The bulk assay shown in Figure 6 may be subject to effects caused by vesicle docking and/or lipid mixing since both effects would result in the observed decrease in donor fluorescence. Content mixing assays should be performed.*

We performed content mixing experiments and confirmed that there was no leakiness in content mixing experiments (Figure 7 and Figure 7—figure supplement 3).

Reviewer #2:

*The role of patches of basic residues at the side (K326, K327) and bottom (R398, R399) of the synaptotagmin 1 C2B domain on SNARE protein/complex and phospholipid binding have been focus of numerous studies in the recent past, leading to partially conflicting results. This study aims to resolve the SNARE and membrane binding modes of the C2B domain (in absence and presence of Ca^2+^) and how these interactions contribute to the fusion process. The authors utilize a number of techniques GST pulldown, coflotation, fluorescence measurements and in vitro fusion assays to demonstrate that mutation of side and bottom basic residues impair SNARE complex binding and membrane binding in a Ca dependent manner. This work is based on a large body of literature and the findings presented here are in part confirmatory and in part new. New is the finding that the interactions of the bottom loop with the membrane can be competitively displaced by the presence of SNARE complexes. Another aspect of the work is that the authors show that synaptotagmin C2B domain is able to double the rate of SNARE complex formation. Perhaps the most interesting aspect is that they can simultaneously monitor both penetration of C2B domain into the membrane and binding to the SNARE complex with the bottom loop at the same time, providing decent proof for the presence of a simultaneous binding of top loops to membranes and bottom loop to cis-SNARE complex via the acid patch created by SNAP25 residues. However, does this experiment exclude that a considerable fraction of the bottom loop would also be able to interact with the phospholipid membrane ?*

We thank reviewer #2 for raising this question. It is generally believed that one synaptic vesicle contains 15 copies of Syt1 molecule. Are all of these Syt1 molecules involved in binding with the SNARE complex? It is really hard to answer based on our bulk assay performed in vitro. However, we found that Syt1-dependent membrane clustering in the presence of calcium was much reduced in the presence of the SNARE complex, we also showed that the C2B2KE or C2B2KA mutations but not the C2B2RQ abolished PIP2 binding with co-flotation experiments. Furthermore, we detected the strong FRET between the bottom loop and the SNARE complex with or without Ca^2+^ in a membrane environment. These results strongly suggest that a majority fraction of the C2B bottom loop interact with the SNARE complex. To data, we could not exclude the possibility that a fraction of the bottom loop bind membranes to promote release. We discussed this possibility in the Discussion (paragraph four). Nevertheless, further work will be required to answer this question by using single molecule FRET experiments.

*On other issue here is the conversion of positive to negative charges in the mutations: net charge difference particularly in the coflotation assays may be of* concern.

We re-tested both the C2B2KE and C2B2KA mutants and the results showed that they both lost the ability to bind PIP2-containing liposomes (Figure 2). Thus, we suppose that conversion of positive charges to negative or neutral charges results in similar effect. Therefore, we used the C2B2KE mutation throughout our study.

Reviewer #3:

*This study has important new data, is largely well-written and, in certain respects, presents major advances, especially in Figure 3. 1) They "propose that the membrane bending induced by the fast transition between the two binding modes underlies the [fusion] triggering function of C2B" (from the Abstract), yet there is no measurement of membrane bending nor is there any data which implicates the transition between the two binding modes, as opposed to the binding modes themselves, in causing fusion. They are welcome to make such a model, of course, but should make clearer distinctions between what they show and what is simply hypothesis.*

Sorry for the misleading sentences in the Abstract or elsewhere in our previous manuscript. In the new one, we described our findings and our hypothesis more clearly. Briefly, our findings are i) before Ca^2+^ influx, C2B binds to the SNARE complex via the bottom R398 R399 region, and interacts with acidic membranes (containing PIP2) through the side K326 K327 region (Figure 2); ii) after Ca^2+^ influx, the R398 R399–SNARE complex interaction and the K326 K327–PIP2 interaction persist upon insertion of the Ca^2+^-binding loops into membranes (Figure 4); iii) we showed that the three binding interfaces of C2B involving the Ca^2+^-binding loops, the R398 R399 and the K326 K327 regions bound respectively to PS, PIP2 and SNARE complex in a simultaneous and synergistic manner (Figure 3 and Figure 4). These interactions were found to be crucial for the triggering activity of Syt1 in liposome fusion (Figure 7). Our hypothesis is that the simultaneous SNARE-containing membrane interactions of the Ca^2+^-binding loops and of the two basic regions of C2B would exert a force to deform the plasma membrane in response to Ca^2+^ (Figure 8). This hypothesis is in good agreement with a recent model proposed by Zhou et al., 2015, Nature. In addition, this hypothesis is supported by a recent observation (Bharat et al., 2014) that local membrane protrusions (5 nm in height, similar like the size of one C2B molecule) bucked on the surface of GUV require the presence of Syt1 and assembled SNARE complex.

*2) The data in Figure 6 from the classical assay of dequenching between two fluorescent lipids purports to show "lipid mixing". However, it has been shown in this journal (Zick and Wickner, eLife 2014;3:e09251) that this assay measures trans-SNARE pair dependent events which do not correspond to lipid mixing. In the text, they assume that this lipid mixing is the result of fusion, though it was shown in this same paper that such dequenching can occur in the absence of fusion. The authors should replace this figure with one measuring the mixing of proteoliposomal lumenal contents (in the continuous presence of an external quenching agent, to control for leakage or lysis).*

We performed content mixing experiments and confirmed that there was no leakiness in our content mixing experiments (see Figure 7 and Figure 7—figure supplement 3). We thank the reviewer for suggesting this robust assay (Zick and Wickner, *eLife* 2014) that has been widely used to measure content mixing. We will definitely try to apply this assay to our future studies.

*3) They claim that calcium binding "…causes a cooperative interaction of C2B with membranes and the SNARE complex." The data however show that C2B can bind membranes or SNARE complex independently, with no data showing cooperativity to this binding.*

We showed that C2B can bind membranes or SNARE complexes independently in the absence of Ca^2+^ (Figure 2), and showed C2B can bind synergistically (Figure 3) and simultaneously (Figure 4) to membranes and SNARE complexes in the presence of Ca^2+^. We agree with reviewer #3 that we had no enough data to declare the “cooperative” behavior among C2B, SNARE complexes and membranes, we thus changed “cooperativity” to “synergistically” throughout the manuscript.

*4) Figure 2; they claim that C2B but not C2A binds to SNARE complex, but the C2B band is faint and the C2A band could be just below the detection limit. Also, they claim synergy between the 2 basic regions of C2B for their binding to SNARE complex, but there is no quantitative binding data to gird this claim. What do they mean by synergy here?*

Our GST pull-down experiments showed that, compared to C2A, either C2B or C2AB exhibited stronger binding to the SNARE complex. The binding between C2A and the SNARE complex might be too weak to be detected in our GST pull-down assay (we noted this in the paper). However, we agree that we could not exclude the possibility that C2A also interacts with the SNARE complex and plays a role in release. Besides, we deleted the sentence claiming the binding synergy of the two basic regions of C2B with the SNARE complex.

*5) Does "MUN", as used in Figure 3, refer to both Munc13 and Munc18? Were these singly omitted, to support the claim that both are needed? I don't see this in the figure.*

As editors suggested, we removed the Munc13 part and only focused on the Syt1–SNARE complex–membrane interaction in the current manuscript.

[Editors' note: the author responses to the re-review follow.]

Essential revisions:

1) The pull-down assay shown in Figure 2 is used as a basis to claim that mutation of the acidic patch in the SNARE complex (SNAP-25 D51/E52/E55 to A – 3M) abolishes binding to the Syt-1 C2B domain. However, it looks as if there is much less SNARE complex in this assay. The authors claim that this is due to lower SDS-resistance of the 3M complex, but there is no trace of the SNARE monomers on the gel, and there is a (slight) difference in the assembly between the WT and the 3M complex. How can one be sure that the same amount of assembled SNARE complex was added in the assay?

We analyzed the overnight-assembled GST-tagged SNARE complex WT and 3M by SDS-PAGE (Figure 2—figure supplement 1). The result show that the SNARE complex 3M is less resistant to SDS compared to the SNARE complex WT before boiling, whereas the individual SNAREs of 3M are same to that of WT after boiling. These results confirm the same amount of assembled SNARE complex was added in our experiments.

*2) Figure 3: In their binding assays designed to show the synergistic effects of the three key Syt-1 C2B binding modes, the authors used a condition where the dominant effect of the calcium-dependent interaction is reduced. How sensitive are the interactions to calcium (see also next point)?*

Previous Figure 3 is current Figure 5; and previous Figure 4 is current Figure 6. As shown in Figure 5—figure supplement 1, we monitored the sensitivity of the binding between C2B and PS-containing liposomes in different Ca^2+^ concentrations. Binding of C2B with liposomes decreased as the Ca^2+^ concentration declined. When the concentration of Ca^2+^ was below 0.1 mM, we could not detect the binding between C2B and PS-containing liposomes with SDS-PAGE gel.

3) Figure 3 implies that there is no interaction between C2B and the membrane-reconstituted SNARE complex in the absence of calcium and absence of PIP2. Yet, Figure 4 suggests that there is such an interaction in the absence of calcium and presence of PIP2. The authors should attempt a co-flotation experiment using the same conditions as that of Figure 4, and in the absence of calcium.

To reduce the dominate effect of the Ca^2+^-dependent interaction between C2B and PS and for better monitoring the synergistic effects among the three key Syt-1 C2B binding modes, we used the buffer with higher ion strength (250 mM KCl) and 0.1 mM Ca^2+^ throughout the experiments in Figure 5 and Figure 5—figure supplement 1. In contrast, the buffer we used for those experiments in Figure 6 and other figures contains 150 mM KCl instead (we noted this in subsection “Ca2+-dependent interactions of C2B to SNARE complexes and acidic membranes”). The salt concentration differences (250 mM v.s. 150 mM) may explain why the C2B–SNARE complex binding could be detected in Figure 6 but not in Figure 5.

Nevertheless, we still performed the co-flotation experiment the reviewers suggested to confirm the interaction between C2B and the membrane-reconstituted SNARE complex using the same conditions as that of Figure 6 (in the presence of 150 mM KCl with and without 0.1 mM Ca^2+^). Please see the figure below; note we did not include this figure in the manuscript).

In this figure, fractions 1 to 6 represent proteins distributed from the top (C2B bound to liposomes) to the bottom (C2B in the pellet) of the density gradients of the co-flotation sample after centrifuge. Liposome compositions are indicated.

Consistent with our prediction, these results show C2B binds to the membrane-embedded SNARE complex but not to plain liposomes in the absence of Ca^2+^ (0.2 mM EDTA) and in the presence of 0.1 mM Ca^2+^, confirming the C2B–SNARE complex interaction in the presence of 150 mM KCl observed in Figure 6.

4) Figure 4: Syt1-T285W and SNAP-25 R59C mutations. The authors should comment in the text that these residues are outside the primary interface between Syt1 and SNARE complex as observed in the crystal structures by Zhou et al. (2015). Thus, these mutations are unlikely going to affect this particular interaction between Syt1 and SNARE complex. This is an important comment since unfortunate placement of labels have affected this interaction in work by others (Brewer et al., 2015; Park et al., 2015). Moreover, the Ca-Ca distance between the two labeling sites is ~ 12 A (based on the structures by Zhou et al.), consistent with the relatively large effect on FRET that the authors observe. Again, this is worth mentioning.

We thank the reviewers for providing these useful comments. We added them in the text (Discussion section paragraph three).

5) Figure 6: Was this experiment done in the presence of membrane (i.e., similar to the method shown in Figure 4)?

Previous Figure 6 is current Figure 3. By using the bimane-tryptophan quenching assay, we performed the experiments of Figure 3 to confirm the maintenance of the R398 R399–SNARE complex interaction in the absence of membranes and in the presence of ATP and Mg^2+^. To avoid the confusion, we change the word “in solution” with “in the absence of membranes”.

6) Figure 7 and second paragraph subsection “Ca^2+^-dependently simultaneous C2B–SNARE complex–membrane interactions are crucial for triggering fusion”: The in vitro assay by Kyoung et al., 2011; Diao et al., 2012; Lai et al., 2014) actually reconstituted the full-length Syt1-SNARE machinery at physiological Syt1 copy number and also included complexin. Moreover, Zhou et al. (2015) showed that mutation of the primary interface disrupts calcium-triggered vesicle fusion in this assay. These results should be mentioned in the context of presenting Figure 7 and the authors should point out which aspects of their experiments go beyond the previous results by Zhou et al. (2015) (e.g., the studies of calcium-binding site mutations.

We mentioned these papers and pointed out the aspects that go beyond the previous results (Results section).

7) Figure 7: Based on other work (Wang Z, Liu H, Gu Y, Chapman ER (2011) Reconstituted synaptotagmin I mediates vesicle docking, priming, and fusion. J Cell Biol 195:1159-70. doi: 10.1083/jcb.201104079) the behaviors of the isolated Syt-1 C2B or Syt-1 C2AC2B fragments are different from the behavior of full-length Syt-1. The authors need to validate the findings in Figure 7 that go beyond those of Zhou et al. (2015) by using full-length, reconstituted Syt-1.

We have validated the functional difference of Syt1 C2A and C2B domain using reconstituted full-length Syt1 (Figure 7—figure supplement 4, and in the final paragraph of the Results section).

8) Figure 8: the palmitoylation positions of the SNAP-25 linker region are incorrectly drawn – they are close to the C-terminal side of the SNARE complex.

We thank the reviewers for pointing out this mistake. We revised it accordingly (Figure 8).

*9) Please provide the rationale for the (relative) concentrations of SNAREs, SNARE complexes, and Syt-1 fragments used in the different assays.*

All SNAREs or SNARE complex-reconstituted liposomes were prepared with a protein-to-lipid ratio of 1:200 as reported (Lai et al., 2014, *eLife*). For GST pull-down and liposome co-flotation assays, excess of the Syt1 fragments (10 μM) were used in order to saturate the binding. For lipid mixing, content mixing (Figure 7) and SNARE paring assays (Figure 7—figure supplement 2), equivalent amount of the Syt1 fragments (0.5 μM) compared to syntaxin-1–SNAP-25 complex (0.5 μM) reconstituted on liposomes were used to avoid liposome cluster during calcium incorporation. For fluorescent measurements, especially FRET-based assays, the Syt1 fragments as a donor (Figure 2) were added to 5 μM in order to monitor the acceptor (rodamine) fluorescence as reported (Hui et al., 2011, NSMB); the Syt1 fragments as an acceptor (for all bimane-tryptophan quenching assays in the manuscript) were added to 5 μM for saturating the FRET, as demonstrated in the titration experiment in Figure 4. For liposome cluster assay in Figure 2—figure supplement 2, excess of Syt1 fragments (10 μM) were used for monitoring liposome cluster as reported (Arac et al., 2006, NSMB and Xue et al., 2008, NSMB). More details are described in the Materials and methods section.

[Editors' note: further revisions were requested prior to acceptance, as described below.]

The manuscript has been improved but there are some small remaining issues that need to be addressed before acceptance, as outlined below:

1) Please move the new results with full-length synaptotagmin^-1^ to the main figure and move the results with the soluble C2AB domain (Figure 7) to a supplement. The results with the full-length protein are more relevant.

We agree with the reviewers that the results with the full-length Syt1 are more relevant and we thus have moved the related results to the main figure (see Figure 8). In addition, we still keep the Figure 7 in the main figures because we think the data obtained in such a more simplified fusion system (with C2B or C2A) is also important for understanding the triggering function of C2B. Both of the C2AB-based and full-length Syt1-based fusion systems well reproduced the functional importance of the C2B domain observed in vivo. Beside, the working model for Syt1 (previous Figure 8) is moved to Figure 9.

2) Results, paragraph one: "K326E/K327E, K326E/K327E" should be "K326A/K327A, K326E/K327E".

We thank the reviewers for careful reading of the manuscript and we revised it as suggested.